# Computational prediction of high-risk non-synonymous SNPs in human ApoE and their structural impact on amyloid-β interaction in Alzheimer's disease pathogenesis

Md. Mainuddin Hossain[ID][1], Juthi Adhikari[2], Amit Dutta[1], Afia Khandaker[1], Sirajul Islam[1], Md Masuder Rahman[1], Abu Zaffar Shibly[ID][1]*

1 Department of Biotechnology and Genetic Engineering, Mawlana Bhashani Science and Technology University, Santosh, Tangail, Bangladesh, 2 Department of Genetic Engineering and Biotechnology, Daffodil International University, Birulia, Savar, Dhaka, Bangladesh

* zaffarshibly@mbstu.ac.bd

## Abstract

Apolipoprotein E (ApoE) plays a critical role in Alzheimer's disease (AD) by regulating amyloid beta (Aβ) clearance through direct interaction. Non-synonymous single nucleotide polymorphisms (nsSNPs) in ApoE alter its structure and impair function, contributing to disease progression. This study aimed to identify functionally damaging nsSNPs in the ApoE gene using *in silico* tools and to assess their structural and binding effects on Aβ in the context of AD progression. A total of 376 nsSNPs were retrieved from dbSNP, ClinVar, and DisGeNET databases. Eight predictive tools (SIFT, PolyPhen-2, PredictSNP, PhD-SNP, PANTHER, PROVEAN, Meta-SNP, and SNAP2) were employed to identify deleterious variants. Protein stability was assessed using I-Mutant 2.0, MUpro, INPS-MD, iStable, and DynaMut2, while structural effects were evaluated via HOPE, MutPred2, Swiss-PDB Viewer, and Missense3D. Domain localization was determined using InterPro. Molecular docking was performed using PyRx and AutoDock Vina. Molecular dynamic simulation (MDS) evaluate binding stability and dynamics through 100-ns by Schrödinger Maestro, analyzing RMSD, RMSF, Rg, SASA. Out of 376 nsSNPs, 10 were consistently predicted as deleterious by all eight computational tools. Among these, two variants (L107P and L122P) were classified as high-risk and located within the receptor-binding domain of ApoE. The receptor-binding domain mediates the interaction between ApoE and Aβ. Molecular docking revealed binding affinities of −5.5 kcal/mol (WT), −5.6 kcal/mol (L107P), and −6.6 kcal/mol (L122P), indicating stronger Aβ binding by L122P. Stronger binding affinity of L122P mutation may promote Aβ aggregation or hinder clearance, potentially contributing to disease severity. MDS showed L122P had the highest structural stability, with the lowest RMSD, RMSF, Rg values and increased SASA, supporting its enhanced interaction with Aβ. These findings suggest

which permits unrestricted use, distribution, and reproduction in any medium, provided the original author and source are credited.

**Data availability statement:** The datasets analyzed during the current study are available in the NCBI repository, accessible at (https://www.ncbi.nlm.nih.gov/snp). FASTA sequence (NP_001289617.1) of ApoE isoform a was retrieved from the NCBI protein database.

**Funding:** The author(s) received no specific funding for this work.

**Competing interests:** The authors have declared that no competing interests exist.

that L122P mutation may enhance Aβ aggregation or hinder its clearance, potentially worsening AD and highlight the structural and functional impact of ApoE variants and the need for experimental validation.

## Introduction

Alzheimer's disease (AD) is a devastating neurodegenerative disorder and the leading cause of dementia, affecting over 55 million individuals globally, a number projected to reach 139 million by 2050 due to an aging population [1,2]. Neuropathologically, it is defined by the extracellular accumulation of amyloid-beta (Aβ) plaques and intracellular neurofibrillary tangles composed of hyperphosphorylated tau protein [3]. The main pathological feature of the disease is the deposition of amyloid β (Aβ) peptides in the brain [4,5] due to dysregulation of Aβ metabolism, particularly impaired clearance [6]. It has been reported that the interaction between apolipoprotein E (ApoE) and Aβ plays a critical role in regulating Aβ aggregation and clearance, thereby directly influencing the formation of amyloid plaques, the dysregulation of Aβ metabolism, and impaired Aβ clearance [7,8].

Genome-wide association studies (GWAS) and sequencing efforts have consistently identified the ApoE gene as the strongest genetic risk factor for late-onset AD (LOAD) [9]. The ApoE gene is located on chromosome 19q13.32 and encodes the apolipoprotein E protein, a 299-amino acid glycoprotein involved in lipid metabolism, synaptic maintenance, and clearance of Aβ peptides from the brain [10,11]. The three major ApoE isoforms ApoE2, ApoE3, and ApoE4 are defined by two single nucleotide polymorphisms (SNPs): rs429358 (C112R) and rs7412 (R158C), which produce distinct amino acid substitutions at positions 112 and 158 [12]. These structural changes profoundly alter ApoE's receptor-binding affinity, lipid-binding efficiency, and interaction with Aβ [13].

ApoE4, defined by arginine at both positions 112 and 158, is present in approximately 15% of the population but accounts for over 50% of AD cases. It is associated with earlier disease onset, increased Aβ aggregation, reduced Aβ clearance, and exacerbated neuroinflammation [14]. In contrast, ApoE2, which carries cysteine at both positions, appears to confer a protective effect, while ApoE3 is considered the neutral or most common isoform [15]. Importantly, the structural and functional differences among these isoforms underscore the critical role of specific amino acid residues in modulating ApoE's interaction with Aβ and its role in AD pathogenesis.

Among genes linked to neurodegeneration, ApoE stands out due to its central role in AD, especially through structural changes introduced by specific non-synonymous single nucleotide polymorphisms (nsSNPs). Beyond the well-characterized isoforms, numerous rare nsSNPs have been identified in ApoE, some of which have been reported in AD patients but remain functionally uncharacterized [16]. Missense mutations in ApoE can perturb receptor binding, lipid transport, and Aβ interactions, functions that are particularly mediated by its receptor-binding domain (residues 1–167), a critical region for interaction with low-density lipoprotein receptors and Aβ peptides [17]. For instance, previously studied variants such as C112R (rs429358) and R158C (rs7412) have been shown to shift

ApoE's domain interactions and affect its lipidation status and aggregation propensity, contributing to AD pathophysiology [18]. Single nucleotide polymorphisms, especially nsSNPs, can exert profound effects on protein conformation, stability, and function. The consequences of these two specific mutations have been investigated; however, many other mutations in the ApoE gene have not yet been studied, and their functional roles remain largely unknown. In particular, the structural implications of these uncharacterized mutations and their potential effects on amyloid-beta (Aβ) binding are poorly understood. Further investigation of these variants may reveal novel insights into the underlying mechanisms of disease.

Identifying deleterious nsSNPs is essential for understanding disease mechanisms and identifying potential biomarkers or therapeutic targets. Nevertheless, experimental characterization of each variant is laborious, time-consuming, and resource-intensive. *In-silico* approaches also offer an efficient and systematic method for prioritizing high-risk mutations by integrating structural, functional, and evolutionary predictions [19]. Given the limitations of experimental approaches, this study employed in silico tools to identify deleterious nsSNPs in ApoE and evaluate their potential structural and functional effects on Aβ interaction.

## Materials and methods

### Retrieving ApoE nsSNPs

Non-synonymous single-nucleotide polymorphisms (nsSNPs) were collected from three comprehensive and widely used databases: the NCBI Short Genetic Variation database (dbSNP) (https://www.ncbi.nlm.nih.gov/snp/) [20], ClinVar (https://www.ncbi.nlm.nih.gov/clinvar/) [21], and DisGeNET (https://disgenet.com/) [22]. We identified and selected nsSNPs from these databases, focusing on coding regions that result in amino acid changes. The dbSNP database offers a comprehensive catalog of short nucleotide variations, encompassing both coding and non-coding regions of the human genome. ClinVar, in contrast, curates variants that have established or suspected associations with clinical phenotypes, supported by empirical evidence. DisGeNET integrates extensive data on gene-disease associations (GDAs) and variant-disease associations (VDAs) derived from multiple biomedical sources. From each database, only missense variants (nsSNPs) were selected for further analysis, as these mutations result in amino acid substitutions that may influence protein structure and function. This study focuses primarily on *in-silico* functional and structural analyses of nsSNPs in the ApoE gene; however, population-level association data, including allele frequencies and genome-wide association study (GWAS) statistics, were not incorporated and represent a limitation of the current work.

Simultaneously, the FASTA sequence (NP_001289617.1) of ApoE isoform a was retrieved from the NCBI protein database (https://www.ncbi.nlm.nih.gov/protein/). We selected NP_001289617.1 for this study, as it encodes a full-length ApoE isoform containing all key functional domains involved in lipid metabolism, receptor binding, and amyloid-beta (Aβ) interaction [10,23]. Compared to the shorter canonical isoform NP_000032.1, it includes additional N-terminal residues that may influence structural integrity and ligand binding [17]. NP_001289617.1, derived from transcript NM_001302688.1, is well-characterized and closely linked to ApoE4-associated Alzheimer's disease (AD) pathology [14]. Other isoforms (e.g., NP_001289618.1–NP_001289620.1), generated through alternative splicing, often lack essential regions or show limited relevance to AD (https://www.ensembl.org/). Furthermore, NP_001289617.1 demonstrates consistent brain expression and is frequently studied in the context of neurodegeneration [10,14].

### Identifying Common Deleterious nsSNPs in ApoE

We employed eight widely used bioinformatics tools to evaluate the potential functional impact of nsSNPs retrieved from SNP databases. The tools were selected based on their reported accuracy and availability at the time of analysis. These tools include These tools include SIFT (https://sift.bii.a-star.edu.sg/) [24], PolyPhen-2 (http://genetics.bwh.harvard.edu/pph2/) [25], PredictSNP (https://loschmidt.chemi.muni.cz/predictsnp/) [26], PhD-SNP (https://snps.biofold.org/phd-snp/phd-snp.html) [27], PANTHER (https://pantherdb.org/) [28], PROVEAN (https://www.jcvi.org/research/provean) [29], Meta-SNP (https://snps.biofold.org/meta-snp/) [30], and SNAP2 (https://bioinformaticshome.com/db/tool/Snap2) [31]. By integrating multiple computational algorithms (sequence-based vs. structure-based), the reliability of variant classification is

substantially enhanced. Consistent predictions across diverse tools strengthen the evidence supporting the pathogenicity of a given variant, while discordant results may reveal distinct mechanistic interpretations, such as protein destabilization versus impaired receptor binding. This integrative, multi-tool strategy provides a more comprehensive and refined assessment of the functional implications of genetic variants. All the deployed programs in this study were utilized solely for benchmarking purposes and not intended for direct clinical or operational application. A summary of these tools and their respective functionalities is provided in **Table 1**.

The "Input Parameters" column is retained to provide researchers with clear guidance on the type of data required to utilize each prediction tool effectively. Since nsSNP analysis tools vary in their data requirements, some needing genomic coordinates (e.g., SIFT), while others require specific protein sequences or amino acid substitutions (e.g., PROVEAN, PolyPhen-2), this information is essential for accurate application and reproducibility in computational workflows. An nsSNP was considered deleterious/damaging if it was predicted as such by at least one of the employed prediction tools. However, nsSNPs consistently predicted as deleterious or damaging across all eight tools were classified as common deleterious. This stringent selection criterion enhanced the reliability and robustness of our findings.

## Predicting the effects of common deleterious nsSNPs on ApoE protein stability

Assessing the impact of mutations on protein stability and structural integrity is crucial. For this predictive tools I-Mutant 2.0 (https://folding.biofold.org/cgi-bin/i-mutant2.0.cgi) [38], INPS-MD (https://inpsmd.biocomp.unibo.it/) [39], iStable (http://predictor.nchu.edu.tw/iStable/) [40],MUpro (https://mupro.proteomics.ics.uci.edu/) [41], and DynaMut2 (https://biosig.lab.uq.edu.au/dynamut2/) [42] were employed to estimate stability changes resulting from single-point mutations, utilizing existing protein structures or sequences and a support vector machine (SVM) approach. I-MUTANT 2.0 predicts the impact of amino acid substitutions on protein stability using ΔΔG values, with values below 0 kcal/mol indicating a decrease in stability. MUpro applies both support vector machine (SVM) and neural network algorithms for stability prediction, also considering ΔΔG values below 0 kcal/mol as indicative of decreasing effects.

INPS-MD integrates both sequence and structural features to estimate ΔΔG values, with mutations yielding values below –0.5 kcal/mol classified as destabilizing. Similarly, DynaMut2 predicted ΔΔG values to evaluate protein stability

**Table 1. Overview of Computational Tools Used for Predicting Deleterious nsSNPs, Including Their Analytical Approaches, Input Parameters, and Accuracy Rate.**

| Prediction Tool | Analysis Approach | Input Parameters | Selection Criteria | Accuracy Rate | Reference |
|---|---|---|---|---|---|
| SIFT | Sequence homology & evolutionary conservation | Chromosome positions, orientations, alleles, or dbSNP | Score < 0.05, Deleterious | 76.99% | [32] |
| PolyPhen-2 | Sequence, phylogenetic, & structural analysis | Protein identifier & amino acid substitutions | Score > 0.5, Probably damaging | 75.56% | [33] |
| PredictSNP | Consensus classifier integrating multiple tools | Query protein sequence & amino acid variations | Score ≥ 0.5, Deleterious | 82% | [26] |
| PhD-SNP | Machine learning (SVM-Sequence & SVM-Profile) | Query protein sequence & amino acid variations | Probability > 0.5, Deleterious | 78% | [34] |
| PANTHER | Evolutionary conservation & molecular function analysis | Query protein sequence & amino acid variations | PSEC score ≤ −3, Damaging | 79.99% | [35] |
| PROVEAN | Sequence alignment & conservation analysis | Protein query sequence & amino acid variations | Score < −2.5, Deleterious | 79.19% | [36] |
| Meta-SNP | Machine-learning model integrating multiple predictors | Query protein sequence & amino acid variations | Score > 0.5, Deleterious | 79% | [30] |
| SNAP2 | Neural network-based evolutionary & structural analysis | Query protein sequence | Score > 0, Deleterious | 82% | [37] |

and flexibility changes from missense mutations. Mutations with ΔΔG values below –0.5 kcal/mol were deemed unstable. Finally, iStable utilizes SVM-based approaches to assess the effect of mutations on protein stability.

### Assessment of the structural impact of common deleterious nsSNPs on ApoE

Structural impacts of common deleterious nsSNPs on the ApoE protein were analyzed using HOPE tool (https://www3.cmbi.umcn.nl/hope/input/) [43], that analyzes the structural consequences of point mutations in protein sequences. To further investigate the structural impact of common deleterious nsSNPs, we utilized MutPred2 (http://mutpred.mutdb.org/) [44], an online tool designed to predict protein pathogenicity based on alterations in amino acid composition. The amino acid substitution data, and ApoE protein sequence in FASTA format, were input into MutPred2. Predictions were classified as "Pathogenic" for g-score > 0.05 and "Non-pathogenic" for g-score < 0.05. Swiss-PDB Viewer 4.1 (https://spdbv.unil.ch/) [45,46] was employed to perform energy minimization for the 3D structures of the mutated ApoE protein models. Swiss-PDB Viewer facilitates the generation of mutant ApoE protein models corresponding to the specified amino acid substitutions. The 'Mutation Tool' in Swiss-PDB Viewer enables users to explore a library of rotamers for the mutated models and select the most suitable rotamer mutated model. Lastly, Missense3D (https://missense3d.bc.ic.ac.uk/) [47] was also employed to validate and strengthen our findings by predicting structural changes resulting from amino acid substitutions, thereby ensuring the robustness and accuracy of the results. Missense3D produces predictions regarding the structural impact of missense variants on transmembrane protein structure.

### Identification of high-risk nsSNPs within ApoE domains

We utilized the InterPro tool (https://www.ebi.ac.uk/interpro/) [48] to locate nsSNPs within the conserved domains of ApoE. This tool identify protein motifs and domains, thereby elucidating the functional characteristics of proteins based on a comprehensive database of domains, protein families, and functional sites [49].

### Prediction of protein-protein interaction networks

STRING v.11.0 (https://string-db.org/) [50] and Cytoscape (v3.10.3) (https://cytoscape.org/) [51] were employed to predict protein-protein interaction. The STRING database provides a comprehensive evaluation and integration of protein-protein interactions, incorporating data from both physical interaction databases and curated biological pathway resources. It predicts potential interactions between ApoE and other proteins. Cytoscape (v3.10.3) was utilized to visualize the imported protein-protein interaction networks between the human ApoE protein and its associated proteins.

### Homology modelling, validation and Molecular Docking study

We employed an online tool SWISS-MODEL (https://swissmodel.expasy.org/) [52] for 3D homology modelling of both the wild-type (WT) and mutant ApoE proteins. SWISS-MODEL was selected for its automated interface, high-quality template selection, and regularly updated structural database, making it well-suited for modelling proteins with available homologs. While tools like MODELLER offer greater customization, SWISS-MODEL provides an optimal balance of accuracy, speed, and accessibility, aligning well with our study's workflow. It also integrates built-in validation tools critical for assessing the quality of the generated structures [53]. Since no complete crystal structure of ApoE is available, SWISS-MODEL enabled us to construct reliable 3D models for both WT and mutant forms, which were then selected for downstream analysis.

The structural models were validated using multiple tools, including, ProSA-web (https://prosa.services.came.sbg.ac.at/prosa.php) [54], QMEAN (https://swissmodel.expasy.org/qmean/) [55], ERRAT [56], and Ramachandran plot analysis via the SAVES server (https://saves.mbi.ucla.edu/). Additionally, to examine interactions with Aβ, we retrieved the Aβ peptide fragment (residues 12–28), a region known to associate with ApoE—from PubChem (CID: 57339251) and processed it using PyMOL [57].

To identify active and binding sites, we utilized the CASTpFold server (https://cfold.bme.uic.edu/castpfold/) [58], which integrates protein folding and cavity topography information to accurately map pockets, cavities, and channels in protein structures. CASTpFold was applied to both WT and the mutant ApoE variants (L122P and L107P).

For assessing the impact of mutations on Aβ binding affinity, we performed molecular docking using PyRx, an open-source virtual screening platform, with the AutoDock Vina v1.2.0 platform [59,60]. The Aβ fragment and ApoE (WT and mutants) were treated as ligand and macromolecule, respectively. The grid box was centered at X: 18.1270, Y: 39.7190, Z: 52.6633, with dimensions X: 53.4957 Å, Y: 27.6560 Å, Z: 48.9882 Å, ensuring coverage of potential interaction sites. The docking results and ligand–protein interactions were visualized and analyzed using Discovery Studio v4.5.

## Molecular dynamic simulation

A 100-nanosecond (ns) molecular dynamics (MD) simulation was conducted to investigate the binding stability of our selected complexes: WT (ApoE)_Aβ, ApoE (L107P) _Aβ, and ApoE (L122P) _Aβ. The simulations were performed using the Maestro 2020.4 software platform, utilizing the OPLS4 force field, on a Linux operating system. This analysis aimed to evaluate the dynamic behavior and structural stability of the respective protein–ligand interactions [61]. Additionally, the TIP3P water model was employed to solvate the system within a predefined orthorhombic periodic boundary box. To neutralize the system and mimic physiological conditions, a 0.15 M concentration of $Na^+$ and $Cl^-$ ions was introduced. Following solvation, the protein–ligand complex underwent energy minimization for 100 picoseconds (ps). All bonds involving hydrogen atoms were constrained using the SHAKE algorithm, the system could be progressively heated to 300 K [62]. Trajectory snapshots were subsequently recorded at 100-ps intervals throughout the simulation. To evaluate the stability of the simulations for the selected complexes, key structural and dynamic parameters were computed, including radius of gyration (Rg), solvent-accessible surface area (SASA), root mean square deviation (RMSD) [63], root mean square fluctuations (RMSF), and torsion angles [64].

## Results

### Retrieval of nsSNPs of ApoE

We retrieved Non-synonymous single nucleotide polymorphisms (nsSNPs) in the Apolipoprotein E (ApoE) gene from three publicly available databases: NCBI dbSNP, ClinVar, and DisGeNET. A total of 381 missense variants were initially identified from the dbSNP database. During data curation, 5 variants were merged based on sequence redundancy, variant clustering, or updated annotations: rs3200542 was merged into rs7412; rs61228756 and rs630496 were merged into rs429358; and rs11542028 and rs678339 were merged into rs440446. After removing these redundant variants, a total of 376 unique missense variants from dbSNP were retained for further analysis.

From the ClinVar database, we identified 120 ApoE-related missense variants, of which 17 were overlapping variants due to multiple clinical submissions or reannotations. These overlapping variants were rs11542041, rs892532644, rs1434405741, rs121918397, rs429358, rs121918393, rs387906567, rs11083750, rs267606664, rs267606661, rs267606663, rs28931577, rs190853081, rs769452, rs201672011, rs429358, and rs387906567.

In parallel, we retrieved 185 ApoE-associated missense mutations from the DisGeNET database, of which 163 were also overlapping variants due to multiple clinical submissions or reannotations, and lastly, 22 unique missense variants were identified.

The nsSNPs obtained from both ClinVar and DisGeNET were also identified among the mutations retrieved from the dbSNP database. Subsequently, a comprehensive comparison and overlap analysis of the three datasets—dbSNP, ClinVar, and DisGeNET—led to the final selection of 376 unique nsSNPs in the ApoE gene. The intersection of these datasets is illustrated in the Venn diagram (Fig 2A). A schematic figure (Fig 1) presents an overview of the methodological approach.

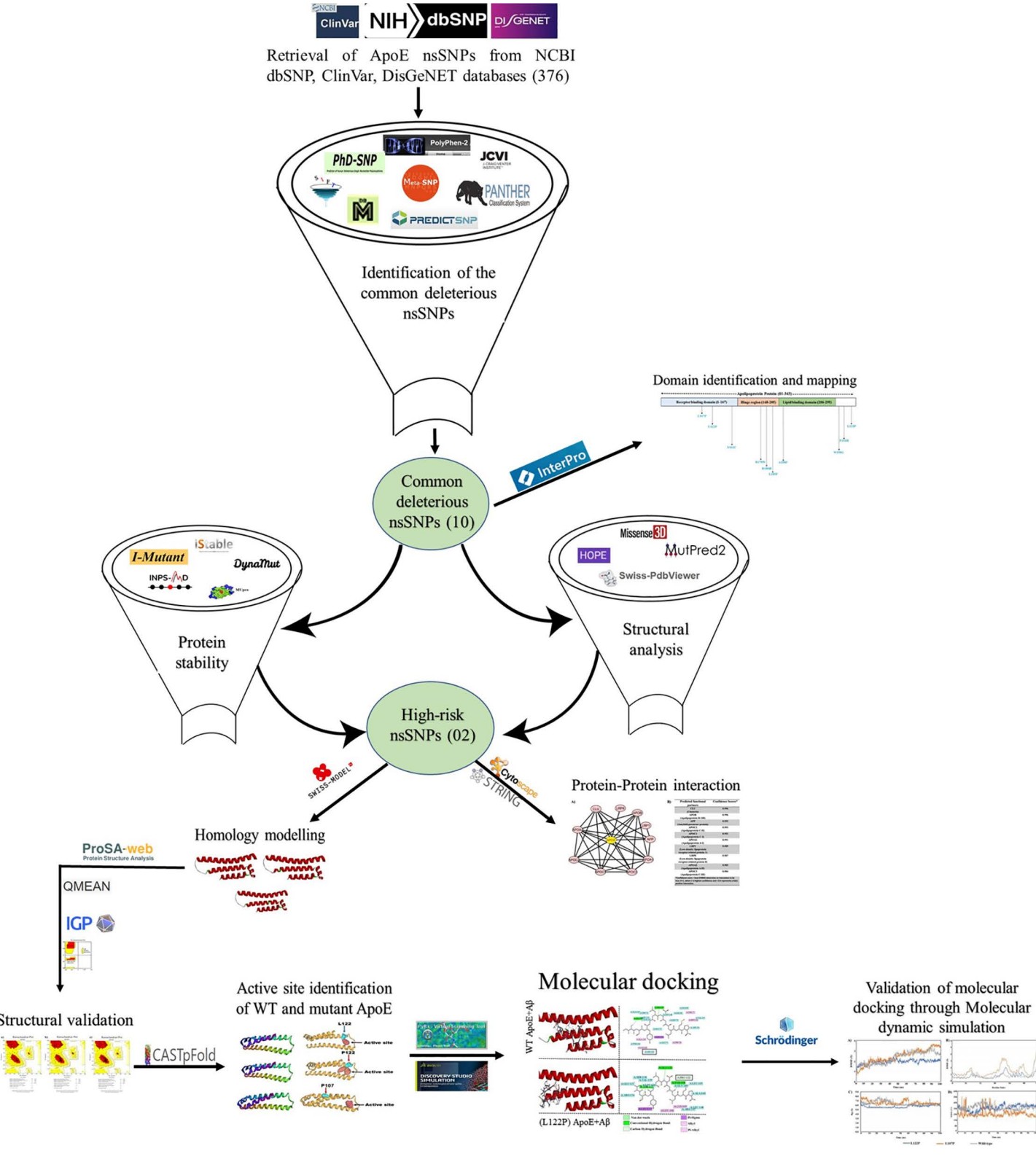

**Fig 1. Schematic representation of the methodological approach.**

## Prediction of common deleterious nsSNPs of ApoE

Firstly, we predict deleterious nsSNPs using eight different *in-silico* prediction tools: SIFT, PolyPhen-2, PredictSNP, PhD-SNP, PANTHER, PROVEAN, Meta-SNP, and SNAP2, which have an effect on the structure or function of the ApoE protein. These tools were selected based on their prediction accuracy and availability at the time of analysis. Out of the 376 nsSNPs analyzed, SIFT identified 178 nsSNPs as deleterious to protein function, with scores of 0.05 or below on a scale from 0 to 1. PolyPhen-2 predicted that 230 nsSNPs would have damaging effects. PolyPhen-2 assigns a probability score ranging from 0.0 (tolerated) to 1.0 (damaging), where values near 0 suggest benign variants, and those near 1.0 indicate likely damaging substitutions.

PredictSNP identified 173 nsSNPs as potentially deleterious. PROVEAN predicted 83 variants with scores below −2.5, a threshold commonly used to indicate a high likelihood of functional damage. Meta-SNP identified 108 nsSNPs as damaging (scores > 0.5), suggesting potential disruption of protein function, particularly in conserved regions. SNAP2 classified 155 variants as deleterious based on a neural network that incorporates evolutionary conservation, predicted secondary structure, and solvent accessibility. PANTHER also predicted 108 nsSNPs to have a functional impact. PhD-SNP identified 31 nsSNPs with high PhyloP100 scores, indicating strong evolutionary conservation at the mutated sites, which implies a greater likelihood of disease association (**Fig 2B**). An nsSNP was classified as common deleterious if all eight computational tools consistently predicted it to be damaging. The term "deleterious/damaging" specifically refers to SNPs predicted to significantly affect protein function, stability, or structure; potentially disease-associated. However, nsSNPs consistently predicted as deleterious or damaging across all eight tools were classified as common deleterious.

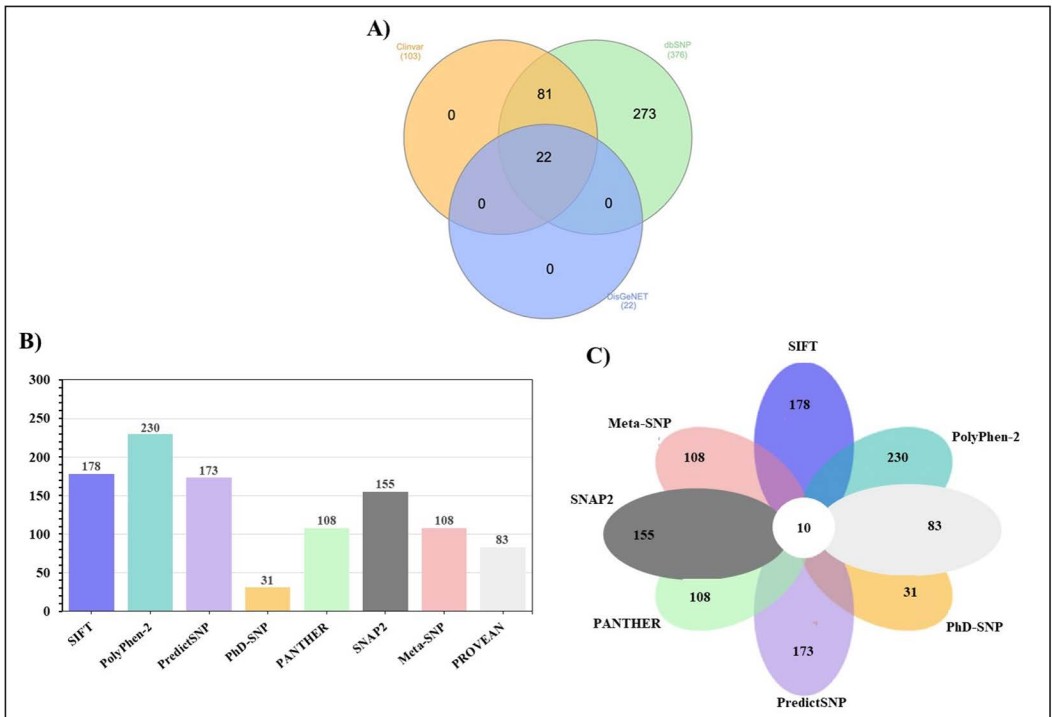

**Fig 2. Data integration and identification of deleterious nsSNPs in the ApoE gene. A)** Venn diagram illustrating the overlap of non-synonymous SNPs (nsSNPs) retrieved from three databases. **B)** A bar chart displays the number of nsSNPs identified by eight [8] different computational tools. **C)** A Venn diagram highlights ten nsSNPs classified as the most deleterious, as they were consistently predicted to be harmful by all eight tools.

Based on this criterion, 10 nsSNPs were identified as common deleterious across all tools (**Table 2**), as illustrated in **Fig 2C** and detailed in S1 Table in S1 File. These common deleterious variants, rs200703101, rs531939919, rs796443813, rs1039600156, rs1238105907, rs1341982092, rs1358158446, rs1457217956, rs1969853804, and rs1969885568, were selected for further in-depth analysis.

## Prediction of the effects of 10 common deleterious nsSNPs on the stability of the ApoE protein

We evaluated the impact of ten common deleterious nsSNPs on ApoE protein stability using five predictive tools: I-MUTANT 2.0, MUpro, INPS-MD, iStable, and DynaMut2. According to I-MUTANT 2.0. nine out of the ten variants (rs200703101, rs531939919, rs796443813, rs1039600156, rs1238105907, rs1341982092, rs1358158446, rs1969853804, and rs1969885568) were predicted to reduce protein stability, exhibiting ΔΔG values below 0 kcal/mol.

Similarly, MUpro predicted all ten nsSNPs to destabilize the ApoE protein, with ΔΔG values less than –0.5 kcal/mol. INPS-MD analysis found that nine variants, except rs531939919, decreased stability, showing ΔΔG values below –0.5 kcal/mol. The iStable tool corroborated these findings, indicating reduced stability in nine variants, excluding rs796443813. DynaMut2 also predicted destabilizing effects for nine nsSNPs, suggesting potential impairment of the protein's structural integrity or functional stability. Notably, rs796443813 was the only mutation predicted to stabilize the protein, as reflected by its ΔΔG value.

The consistent results across all five tools— I-MUTANT 2.0, MUpro, INPS-MD, iStable, and DynaMut2—highlight that these nsSNPs predominantly compromise the structural stability of the ApoE protein, underscoring their potentially deleterious impact (see Table 3).

## Prediction of the effects of 10 common deleterious nsSNPs on the structure of the ApoE protein

Computational tools HOPE, MutPred2, Swiss PDB Viewer, and Missense3D were used to analyse the structural impact of 10 common deleterious nsSNPs on ApoE. Missense3D was employed to assess structural disruptions induced by amino acid substitutions. The analysis identified 6 common deleterious nsSNPs including, rs796443813 (A208P), rs1039600156 (L323P), rs1238105907 (L122P), rs1358158446 (P311H), rs1969853804 (L107P), and rs1969885568 (W308G), were predicted to cause structural damage, including atomic clashes, cavity alterations, the introduction of buried proline residues, and changes to secondary structure. In contrast, no significant structural disruptions were predicted for the rs200703101 (R180H), rs531939919 (R178W), rs1341982092 (Y162C), and rs1457217956 (L185P) variants (**Table 3**). Detailed results were provided in S2 Table in S1 File.

Moreover, MutPred2 server result revealed that rs1238105907 (L122P), rs1039600156 (L323P), rs1969853804 (L107P), and rs1969885568 (W308G) showed g-score greater than 0.5, indicating strong pathogenicity (**Table 3**). The

**Table 2. Common deleterious nsSNPs [10] identified by eight in-silico prediction tools.**

| SNP ID | AA Change | SIFT | Polyphen-2 | PredictSNP | PhD-SNP | PANTHER | SNAP2 | Meta-SNP | PROVEAN |
|---|---|---|---|---|---|---|---|---|---|
| rs200703101 | R180H | 0.02 | 0.897 | Deleterious | Disease | Damaging | Effect [45] | 0.65 | −7.058 |
| rs531939919 | R178W | 0.01 | 0.644 | Deleterious | Disease | Damaging | Effect [12] | 0.68 | −8.349 |
| rs796443813 | A208P | 0.01 | 0.958 | Deleterious | Disease | Damaging | Effect [14] | 0.55 | −5.642 |
| rs1039600156 | L323P | 0.03 | 0.958 | Deleterious | Disease | Damaging | Effect [23] | 0.58 | −3.435 |
| rs1238105907 | L122P | 0.01 | 1 | Deleterious | Disease | Damaging | Effect [45] | 0.78 | −5.775 |
| rs1341982092 | Y162C | 0.04 | 0.823 | Deleterious | Disease | Damaging | Effect [14] | 0.65 | −5.779 |
| rs1358158446 | P311H | 0.02 | 0.653 | Deleterious | Disease | Damaging | Effect [56] | 0.51 | −4.89 |
| rs1457217956 | L185P | 0.04 | 0.958 | Deleterious | Disease | Damaging | Effect [1] | 0.72 | −5.828 |
| rs1969853804 | L107P | 0.01 | 1 | Deleterious | Disease | Damaging | Effect [1] | 0.73 | −4.252 |
| rs1969885568 | W308G | 0.03 | 0.653 | Deleterious | Disease | Damaging | Effect [12] | 1 | −6.219 |

Table 3. Stability and structural analysis of ApoE protein caused by 10 common nsSNPs.

| SNP ID | AA Change | I-MUTANT | | MUpro | | INPS-MD | | iStable | DynaMut2 | | Missense3D | MutPred2 | | Swiss PDB Viewer | HOPE |
|---|---|---|---|---|---|---|---|---|---|---|---|---|---|---|---|
| | | Prediction | ΔΔG value (Kcal/mol) | Prediction | ΔΔG value (Kcal/mol) | Prediction | ΔΔG value (Kcal/mol) | | Prediction | ΔΔG value (Kcal/mol) | | Prediction | ΔΔG value (Kcal/mol) | | |
| rs200703101 | R180H | DS | −1.28 | DS | −1.0834525 | DSS | −0.93 | DS | DSS | −0.79 | NSD | NSA | 0.117 | L0 | X |
| rs531939919 | R178W | DS | −0.09 | DS | −0.71056904 | NS | −0.33 | DS | DSS | −1.86 | NSD | NSA | 0.178 | L0 | X |
| rs796443813 | A208P | DS | −0.83 | DS | −0.71005494 | DSS | −1.16 | IS | SS | 0.2 | CL | NSA | 0.219 | L1 | X |
| rs1039600156 | L323P | DS | −1.32 | DS | −1.7782903 | DSS | −2.75 | DS | DSS | −1.5 | CA | SA | 0.578 | L0 | X |
| rs1238105907 | L122P | DS | −1.19 | DS | −1.6171938 | DSS | −3.11 | DS | DSS | −1.49 | BPI, SSA | SA | 0.797 | L2 | H |
| rs1341982092 | Y162C | DS | −0.14 | DS | −1.1623727 | DSS | −1.53 | DS | DSS | −1.73 | NSD | NSA | 0.239 | L0 | X |
| rs1358158446 | P311H | DS | −0.87 | DS | −1.3664782 | DSS | −0.99 | DS | DSS | −1.26 | SSA | NSA | 0.452 | L0 | X |
| rs1457217956 | L185P | IS | 1.03 | DS | −2.0717421 | DSS | −2.38 | DS | DSS | −1.65 | NSD | NSA | 0.422 | L0 | X |
| rs1969853804 | L107P | DS | −0.83 | DS | −2.5595529 | DSS | −2.05 | DS | DSS | −1.46 | BPI | SA | 0.596 | L2 | H |
| rs1969885568 | W308G | DS | −3.12 | DS | −1.8959951 | DSS | −2.82 | DS | DSS | −3.73 | CA | SA | 0.622 | L0 | X |

Decrease stability: DS, Increase stability: IS, Destabilizing stability: DSS, Stabilizing stability: SS, Neutral stability: NS, No Structural Damage Detected: NSD, Clash: CL, Cavity altered: CA, Buried Proline introduced: BPI, Secondary Structure Altered: SSA, Structure alteration: SA, No structure alteration: NSA, No H₂ bond loss: Lo0, Loss of one H₂ bond: L1, loss of two H₂ bonds: L2, Disruption of H₂ bond: H, No H₂ bond disruption: X.

I-MUTANT (ΔΔG <0 kcal/mol), MUpro (ΔΔG <0 kcal/mol), INPS-MD (ΔΔG <0 kcal/mol), iStable (ΔΔG <0.5 kcal/mol), DynaMut2 (ΔΔG <0 kcal/mol), Missense3D (structural damage detected), MutPred2 (g-score>0.5), Swiss PDB Viewer (loss of hydrogen bond), HOPE (disruption of hydrogen bond).

system also predicted whether these mutations led to any gains in intrinsic disorder. Additionally, the loss of SUMOylation was specifically associated with the rs1238105907 (L122P) mutation. Both rs1238105907 (L122P) and rs1969885568 (W308G) were found to induce similar alterations in the disordered interface region. Additionally, disrupted metal-binding properties were observed in rs1039600156 (L323P) and rs1969885568 (W308G), while altered DNA-binding capacity was also detected in rs1969885568 (W308G). Furthermore, structural changes affecting the transmembrane domain were identified in rs1238105907 (L122P), rs1969853804 (L107P), and rs1969885568 (W308G) (S3 Table in S1 File).

Analysis using the Swiss-PDB Viewer mutation tool revealed that the rs796443813 (A208P) variant resulted in the loss of one hydrogen bond, while rs1238105907 (L122P) and rs1969853804 (L107P) each led to the loss of two hydrogen bonds. These structural disruptions suggest that the rs796443813 (A208P), rs1238105907 (L122P), and rs1969853804 (L107P) variants may induce conformational changes and alter the energetic stability of the ApoE protein (**Fig 3** and **Table 3**).

Lastly, the Hope server analysed the structural impact of common deleterious nsSNPs on a protein sequence, focusing on alterations in amino acid size, charge, hydrophobicity, and spatial configuration. Our results indicated that each amino acid substitution affects the protein's charge and size. Among the 10-common deleterious nsSNPs evaluated, 2 nsSNPs,

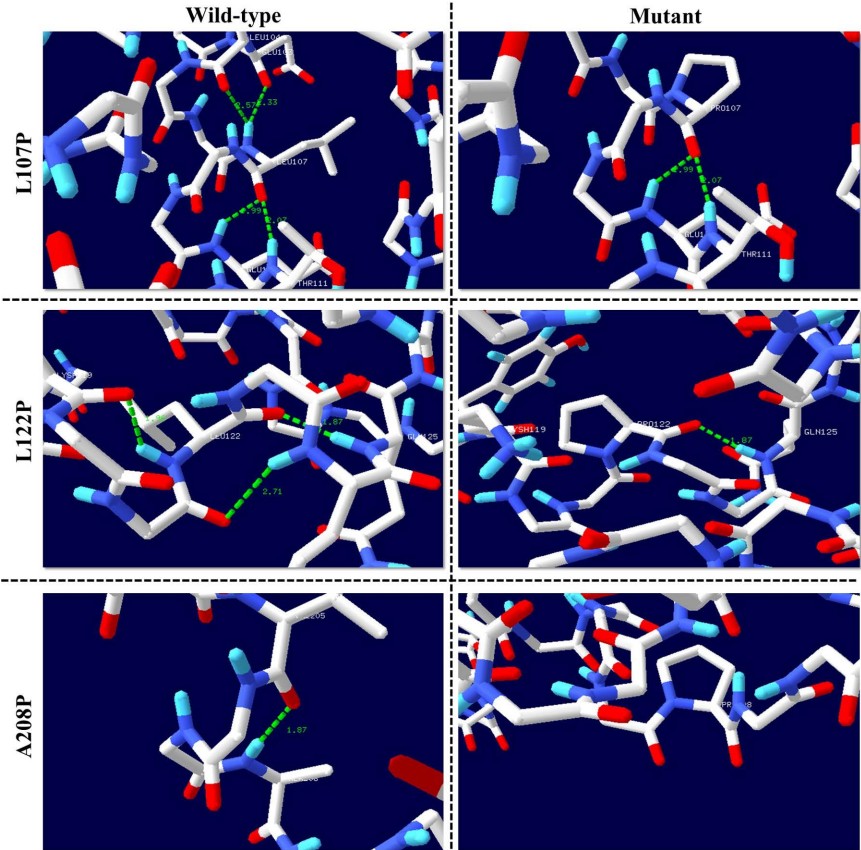

**Fig 3. Structural alterations and hydrogen bonding patterns induced by mutations in the ApoE protein, visualized using Swiss PDB Viewer.** L107P mutation: The wild-type residue forms four hydrogen bonds, while the mutant residue forms only two. L122P mutation: The wild-type residue establishes three hydrogen bonds, reduced to two in the mutant, disrupting local structural stability. A208P mutation: The wild-type residue forms one hydrogen bond, while the mutant residue forms no bond. The 3D representations illustrate the wild-type (green) and mutant (red) residues and their respective hydrogen bond networks (indicated by green dashed lines).

rs1238105907 (L122P) and rs1969853804 (L107P), influenced size, altered charge, and disrupted hydrogen bonding within the ApoE protein (**Table 3**), potentially affecting its interactions with other molecules. A comprehensive results of HOPE server provided in S4 Table in S1 File.

**Table 3** emphasized significant impacts on protein stability, and structural integrity across all employed computational prediction tools, including I-MUTANT, MUpro, INPS-MD, iStable, DynaMut2, Missense3D, MutPred2, Swiss-PDB Viewer, and HOPE. The consistent predictions across multiple computational platforms enhance the reliability of the findings.

### Identification of two high risk nsSNPs

A decrease in protein stability leads to misfolding, degradation, and aggregation, resulting in medical problems [65]. In our protein stability analysis, we found nine common deleterious nsSNPs, with ΔΔG values below −0.5, have been linked to ApoE protein instability, which are rs200703101 (R180H), rs531939919 (R178W), rs796443813 (A208P), rs1039600156 (L323P), rs1238105907 (L122P), rs1341982092 (Y162C), rs1358158446 (P311H), rs1969853804 (L107P), and rs196988.

In contrast, when analysing structural changes in proteins, Missense3D identified six damaging nsSNPs, while Mut-Pred2, Swiss PDB Viewer, and HOPE identified four, three, and two harmful variants, respectively, that altered the structure of the ApoE protein. Among the ten common deleterious nsSNPs, the variants rs1238105907 (L122P) and rs1969853804 (L107P) were consistently predicted to be structurally damaging. These mutations led to significant alterations, including atomic collisions, cavity modifications, the introduction of buried proline residues, and disruptions in the protein's secondary structure. Across eight computational tools, L122P and L107P were consistently classified as high-risk variants due to their pronounced impact on ApoE's structural integrity and stability (Fig 4).

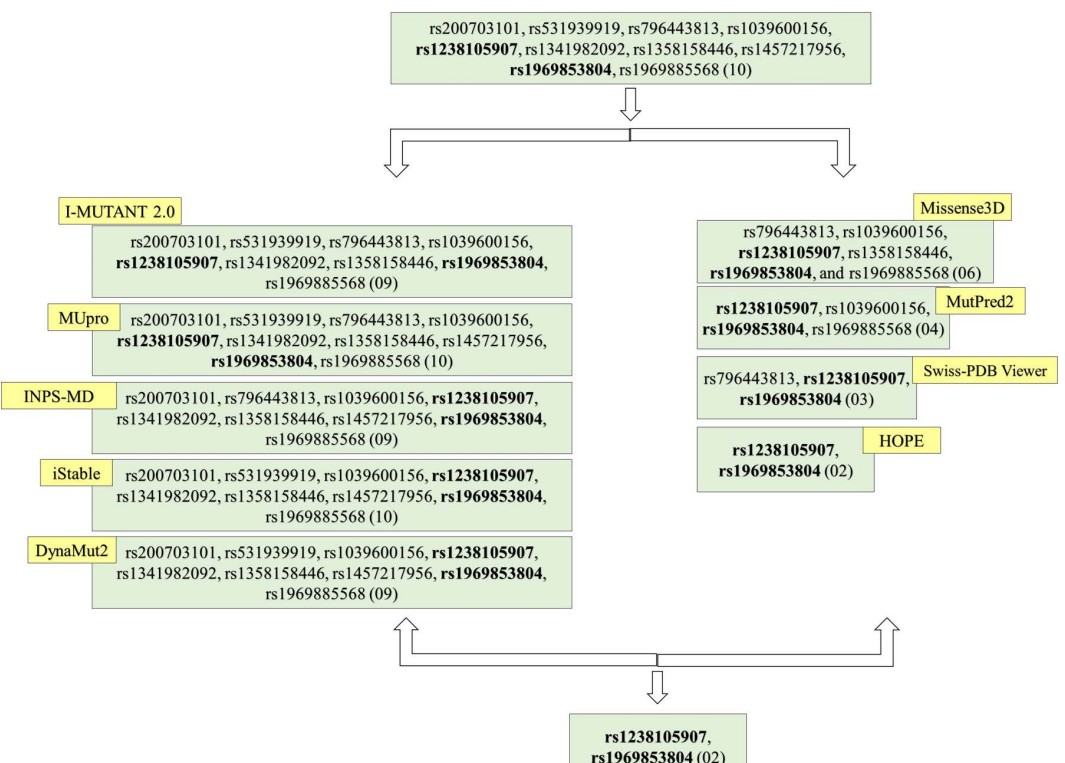

**Fig 4. Selection of high-risk [2] nsSNPs rs1238105907 (L122P) and rs1969853804 (L107P) utilizing computational tools.** Using computational tools, we identified two high-risk genetic variants—rs1238105907 (L122P) and rs1969853804 (L107P)—that are likely to affect both the stability and structure of the protein.

The term "high-risk" specifically refers to variants that significantly affect protein function, stability and structural desta-bilization across all computational prediction tools (e.g., I-MUTANT, MUpro, INPS-MD, iStable, DynaMut2, Missense3D, MutPred2, Swiss-PDB Viewer, and HOPE) suggesting a higher potential relevance to disease pathology.

## Domain Identification of ApoE and Location of High-Risk nsSNPs

Domain identification tool InterPro predicted protein domains and active sites through functional protein family analysis. InterPro identified two functional domains within the ApoE protein: the receptor binding domain (1–167 residues) and the lipid binding domain (206–299 residues), together with a hinge region (168–205 residues). Our analysis identified that 4 common deleterious nsSNPs (L107P, L122P, Y162C, and A208P) are located within functional domains of the ApoE protein. Additionally, 3 nsSNPs (R178W, R180H, and L185P) were situated within the hinge region, while the remaining 3 variants (W308G, P311H, and L323P) were located outside both functional domains and the hinge region those are shown in Fig 5. Notably, the high-risk nsSNPs L107P and L122P were positioned within the receptor-binding domain.

## Prediction of protein-protein interaction

We explored STRING v.11.0 and CytoScape to predict the interaction between ApoE proteins with other pro-teins. We identified that ApoE established a network signaling with Clusterin (CLU), Apolipoprotein B-100 (APOB), Amyloid-precursor protein (APP), Apolipoprotein C-II (APOC2), Apolipoprotein C-I (APOC1), Apolipoprotein A-I (APOA1), Low-density lipoprotein receptor-related protein 1 (LRP1), Low-density lipoprotein receptor-related protein 8 (LRP8), Apo-lipoprotein A-II (APOA2), and Apolipoprotein C-III (APOC3) (Fig 6A), as indicated by their confidence score in (Fig 6B). High confidence scores reflect strong, validated interactions, emphasizing ApoE's central role in AD pathology.

Biologically, the identified interacting partners of ApoE are central to lipid metabolism, amyloid-beta (Aβ) clearance, synaptic function, and neuroinflammation—key processes implicated in AD pathogenesis. For instance, APP is cleaved to produce Aβ peptides, the accumulation of which is a hallmark of AD; ApoE isoforms differentially influence Aβ aggregation and clearance. Clusterin (CLU) and LRP1 are also involved in Aβ transport across the blood-brain barrier, and alterations in ApoE-LRP1 interactions may disrupt this process, leading to impaired Aβ clearance and plaque formation. Similarly,

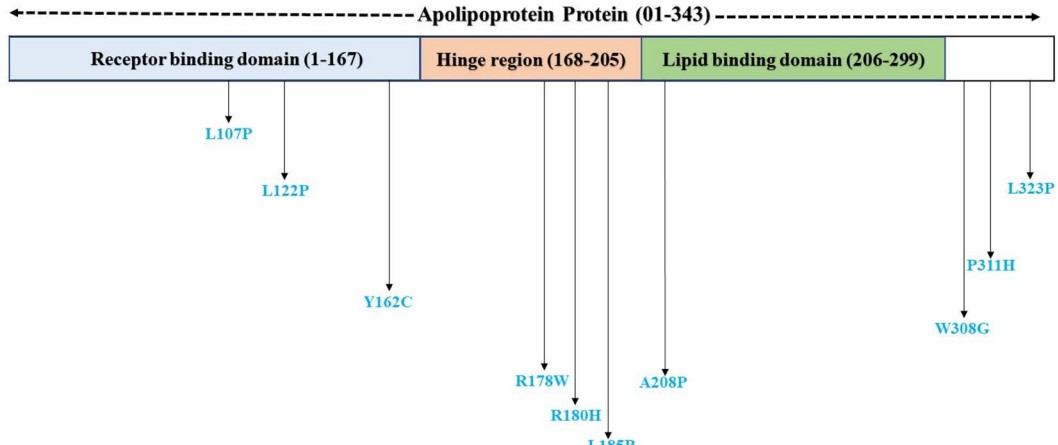

**Fig 5. Domain identification of ApoE protein using InterPro server.** It predicted the receptor binding domain (1-167), lipid binding domain (206-299) and hinge region (168-205) of ApoE.

A)

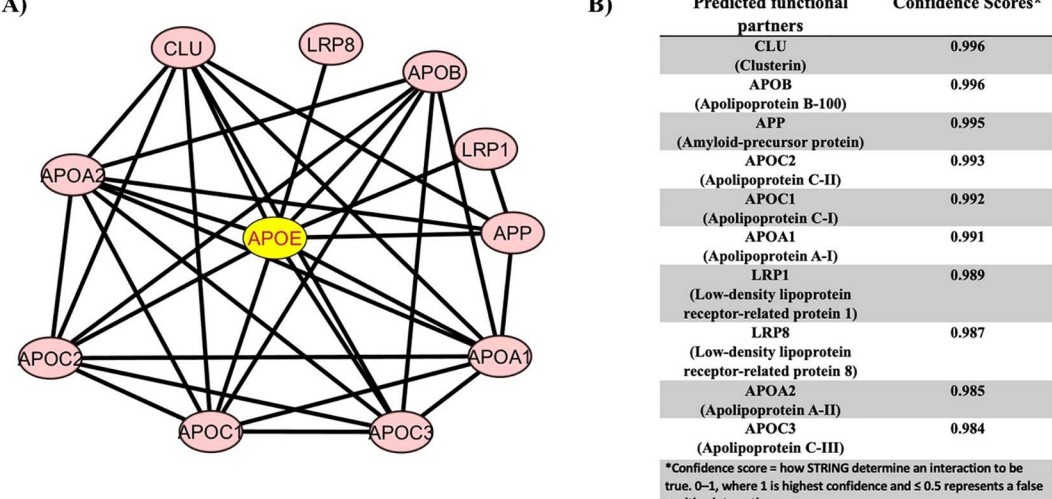

B)

| Predicted functional partners | Confidence Scores* |
|---|---|
| CLU (Clusterin) | 0.996 |
| APOB (Apolipoprotein B-100) | 0.996 |
| APP (Amyloid-precursor protein) | 0.995 |
| APOC2 (Apolipoprotein C-II) | 0.993 |
| APOC1 (Apolipoprotein C-I) | 0.992 |
| APOA1 (Apolipoprotein A-I) | 0.991 |
| LRP1 (Low-density lipoprotein receptor-related protein 1) | 0.989 |
| LRP8 (Low-density lipoprotein receptor-related protein 8) | 0.987 |
| APOA2 (Apolipoprotein A-II) | 0.985 |
| APOC3 (Apolipoprotein C-III) | 0.984 |
| *Confidence score = how STRING determine an interaction to be true. 0–1, where 1 is highest confidence and ≤ 0.5 represents a false positive interaction. | |

**Fig 6. Prediction of protein–protein interactions (PPI) using STRING v. 11.0 and Cytoscape. (A)** Protein–protein interaction network of ApoE (yellow node) with key proteins involved in lipid metabolism and Alzheimer's disease, including CLU, LRP8, APOB, LRP1, and APP. These interactions are crucial for processes like amyloid beta (Aβ) clearance, lipid transport, and neuroinflammation in AD. **(B)** Confidence scores of the interactions indicating the reliability of these connections based on experimental and computational data.

APOA1, APOC1, APOC2, and APOC3 are involved in lipid transport and redistribution, functions that are essential for maintaining neuronal membrane integrity and synaptic plasticity.

The network further highlights ApoE's role as a central hub protein in regulating signaling pathways and homeostatic mechanisms critical for brain development and function. Any structural or functional alteration in ApoE—such as those caused by high-risk nsSNPs (L107P and L122P) may disrupt its interactions with these proteins, thereby perturbing lipid metabolism, impairing Aβ processing, and promoting neuroinflammation. This disruption could facilitate the cascade of molecular events that ultimately contribute to neurodegeneration and the development of Alzheimer's disease.

Therefore, the PPI findings not only validate the functional significance of ApoE within a broader signaling network but also provide mechanistic insights into how genetic variations in ApoE may drive pathological processes in AD. These interactions may serve as potential targets for therapeutic intervention aimed at restoring network homeostasis in the context of disease.

## Homology modelling and validation

We employed the SWISS-MODEL tool to generate 3D structures for both WT and high-risk mutant ApoE proteins [PDB ID (1B68)]. We selected the top model for both WT and the high-risk mutant ApoE from SWISS-MODEL for further analysis. The modelled ApoE sequence corresponds to the ApoE4 isoform, which is defined by the presence of arginine at both positions 112 and 158, determined by the SNPs rs429358 and rs7412. This isoform is the most significant genetic risk factor for late-onset Alzheimer's disease and is known to alter ApoE structure, lipid metabolism, and amyloid-beta (Aβ) interaction dynamics. We then validated these structures using ProSA-web, QMEAN, ERRAT, and Ramachandran plot analysis. The validation checks on the 3D models showed that both the normal and high-risk mutant versions were of good quality, according to the ERRAT server. The quality factors were 96.9027 for the wild-type model, 94.3218 for the rs1238105907 (L122P) variant, and 95.5984 for rs1969853804 (L107P). The structural quality was also backed up by ProSA-Z scores and QMEAN Z-scores, which showed that the models were reliable. The Ramachandran plot analysis (S1 Fig in S1 File) indicated that most of the amino acids were in stable areas, with 93.7% for the wild-type, 92.4% for rs1238105907 (L122P), and 93.2% for rs1969853804 (L107P). (Table 4). Collectively, the structural validation metrics suggest that the L122P and L107P mutations

**Table 4. Structural Validation of Wild-Type and High-Risk ApoE Mutant Protein Models.**

| SNP ID | AA Change | Residues in most favored regions (Ramachandran plot) | Overall quality factor (ERRAT) | ProSA Z-score | QMEAN Z-Score |
|---|---|---|---|---|---|
| | Wild-type | 208 (93.7%) | 96.9027 | −3.88 | −7.61 |
| rs1238105907 | L122P | 219 (92.4%) | 94.3218 | −3.86 | −7.19 |
| rs1969853804 | L107P | 218 (93.2%) | 95.5984 | −3.67 | −7.84 |

induce minor reductions in overall structural quality and conformational stability relative to the wild-type ApoE. Nevertheless, the predicted models remain within acceptable validation thresholds, thereby supporting their suitability for downstream computational analyses, including molecular docking and molecular dynamics simulations.

## Analysis of molecular docking

To evaluate the potential functional consequences of high-risk ApoE mutations in the context of AD, molecular docking was employed to assess the interaction between the wild-type (WT) and mutant ApoE proteins (L122P and L107P) with amyloid-beta (Aβ) peptides. This approach was critical for predicting changes in binding affinity and identifying specific amino acid residues involved in Aβ recognition, which are pivotal in Aβ aggregation, clearance, and neurotoxicity, key hallmarks of AD pathophysiology.

The CASTpFold analysis revealed significant alterations in the active site topology following mutation. The L122P mutation notably expanded the active site surface area from 70.701 Å² (WT) to 101.561 Å², whereas L107P caused only a marginal increase (70.996 Å²). The marked enlargement and visible repositioning of the active site cavity in the L122P variant (**Fig 7B**) suggest substitution of leucine with proline at position 122 introduces a conformational shift that modifies the accessibility and chemical environment of the binding pocket. This is consistent with proline's known structural rigidity, which can disrupt local helicity and induce kinks or loops, thereby altering spatial orientation of neighboring residues. A comprehensive result of CASTpFold analysis were given S5 Table in S1 File.

Molecular docking results support these structural changes, demonstrating that L122P significantly increased binding affinity with Aβ (–6.6 kcal/mol) compared to both the WT (–5.5 kcal/mol) and L107P (–5.6 kcal/mol) proteins (**Table 5**). Notably, the wild-type residue Leu122 did not directly participate in Aβ binding (**Fig 8A**). However, in the mutant structure, **Pro122** engaged in Van der Waals interactions, while neighboring residues such as Tyr118, Gln125, and Thr127 formed new hydrogen bonds, reinforcing the idea that the mutation reshapes the local binding environment to facilitate tighter Aβ engagement (**Fig 8B**).

This increased binding potential has functional implications: stronger ApoE (L122P)–Aβ interactions represent a gain-of-function effect that could promote Aβ aggregation through decrease Aβ clearance, and increase deposition in the brain exacerbating AD pathology.

## Molecular dynamic simulation

Molecular dynamics (MD) simulation was employed to assess the stability and binding behavior of ApoE–Aβ complexes under physiologically relevant conditions. This approach enables the observation of real-time conformational changes and interaction dynamics, simulating a biological environment to validate protein–ligand interactions. To evaluate the impact of high-risk mutations on ApoE–Aβ binding stability, 100 ns simulations were conducted for the wild-type ApoE, two mutant variants (L107P and L122P), and their respective complexes with Aβ. The analysis aimed to identify structural deviations, interaction persistence, and overall complex stability, thereby providing mechanistic insights into how these mutations influence Aβ association and potentially contribute to AD pathogenesis.

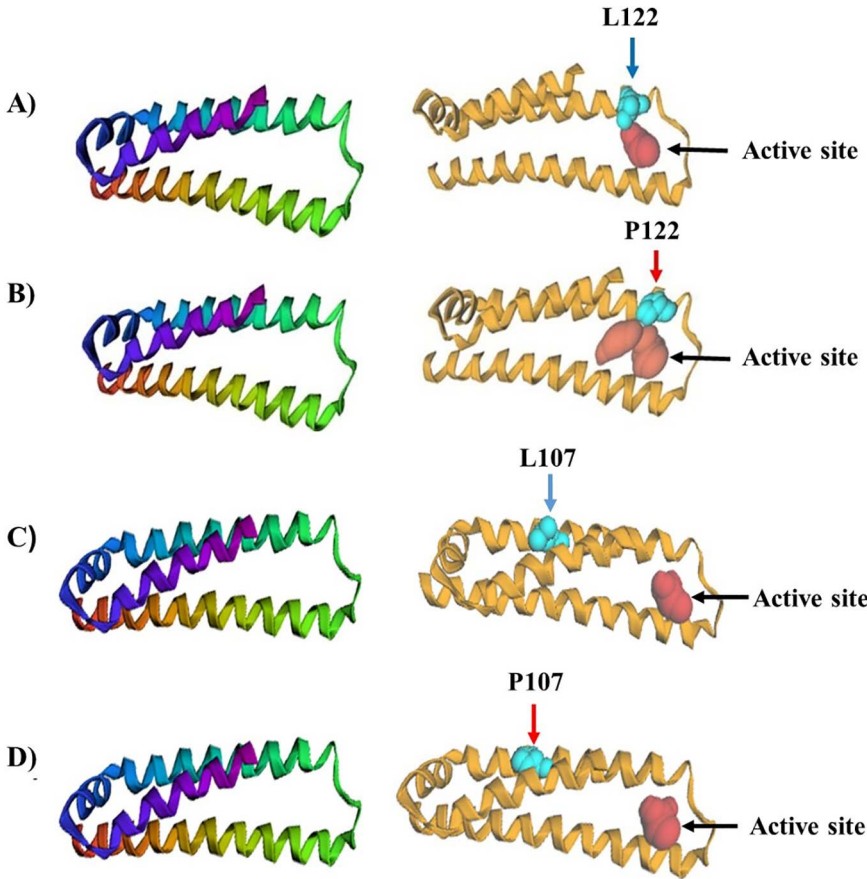

**Fig 7. Structural comparison of the active sites in wild-type and mutant ApoE protein models generated by the CASTpFold server.** The ribbon diagrams show the structure of ApoE in four forms: (A) the WT version with Leucine at position 122, (B) the mutated version with Proline at position 122, (C) the WT version with Leucine at position 107, and (D) the mutated version with Proline at position 107. In each model, the red spherical surface marks the active site's surface area.

The Root Mean Square Deviation (RMSD) profiles of the wild-type, L107P, and L122P protein variants were analyzed over a 100 ns molecular dynamics simulation to evaluate their structural stability. Typically, an RMSD range of 1–3 Å is considered acceptable for a stable protein-ligand complex during simulation [66]. The L107P mutant exhibited a progressive increase in RMSD, reaching values close to 3.5–4.0 Å, indicating significant conformational instability and structural deviation from the initial configuration. The wild-type structure fluctuated within the 2.0–3.0 Å range, suggesting moderate stability. In contrast, the L122P mutant consistently maintained the lowest RMSD values, fluctuating within 1.5–2.5 Å, which is indicative of enhanced conformational stability. These findings suggest that the L122P mutation may contribute to a more structurally stable protein, whereas the L107P mutation destabilizes the protein structure (**Fig 9A**).

Root Mean Square Fluctuation (RMSF) analysis was performed to assess the flexibility of individual amino acid residues. Generally, RMSF values below 2.0 Å represent a stable protein with minimal residue-level fluctuations, whereas values exceeding 2.5 Å indicate highly flexible or disordered regions [67]. The L107P mutant exhibited significantly elevated RMSF values, particularly within the residue index range 48–80, reaching peaks of 5.0 Å, which suggests increased local flexibility and potential destabilization of critical structural regions. The wild-type protein showed moderate fluctuations, with most residues remaining within 1.0–2.5 Å. Conversely, the L122P mutant demonstrated consistently lower

**Table 5. Binding affinities and contact residues of WT and mutant ApoE protein with Aβ.**

| Receptor | Ligand | Binding Affinity (kcal/mol) | Contact residues | Bonding type |
|---|---|---|---|---|
| ApoE (WT) | Aβ | −5.5 | PHE77 | Alkyl |
| | | | TRP78 | Alkyl |
| | | | LYS119 | Alkyl |
| | | | GLU140 | Conventional Hydrogen Bond |
| | | | LEU141 | Pi-sigma |
| | | | ALA144 | Alkyl |
| | | | ARG147 | Conventional Hydrogen Bond |
| | | | MET152 | Alkyl |
| ApoE (L107P) | Aβ | −5.6 | ARG82 | Conventional Hydrogen Bond |
| | | | GLN85 | Conventional Hydrogen Bond |
| | | | LEU148 | Alkyl |
| | | | ASP151 | Conventional Hydrogen Bond |
| | | | VAL155 | Unfavorable donor-donor bond |
| | | | TYR162 | Pi-alkyl |
| **ApoE (L122P)** | **Aβ** | **−6.6** | TYR118 | Conventional Hydrogen Bond |
| | | | LYS119 | Alkyl |
| | | | **PRO122** | **Van der Waals** |
| | | | GLN125 | Conventional Hydrogen Bond |
| | | | THR127 | Conventional Hydrogen Bond |
| | | | LEU137 | Pi-sigma |
| | | | LEU141 | Alkyl |

fluctuations across the structure, with RMSF values largely remaining below 1.5 Å, indicating increased rigidity and a more stable conformation (**Fig 9B**).

The Radius of Gyration (Rg) provides insight into the overall compactness of the protein structure. For a globular protein of this size, Rg values typically range from 5.0 to 7.0 Å [68]. The L107P mutant displayed persistently higher Rg values, fluctuating between 6.0–8.0 Å, which suggests a loosely packed and expanded structure, likely due to conformational instability. The wild-type protein maintained Rg values around 6.0–7.0 Å, indicating a moderately compact structure. In contrast, the L122P mutant consistently exhibited the lowest Rg values, stabilizing near 5.0–6.0 Å, indicative of a tightly folded, compact structure, which correlates with its lower RMSD and RMSF profiles (**Fig 9C**).

Solvent Accessible Surface Area (SASA) analysis measures the extent of the protein's surface exposed to the solvent, which is typically associated with protein folding and stability. Stable globular proteins generally exhibit SASA values within 300–500 Å² [69]. The L107P mutant displayed markedly lower SASA values, fluctuating predominantly between 150–350 Å², suggesting reduced solvent exposure and possible structural collapse or misfolding. The wild-type protein exhibited intermediate SASA values ranging from 300–450 Å², reflecting moderate solvent accessibility. The L122P mutant, however, maintained consistently higher SASA values within the 350–500 Å² range, indicative of a well-folded structure with favorable solvent exposure, supporting its overall structural stability (**Fig 9D**).

A 100 ns molecular dynamics simulation was performed to evaluate intermolecular interactions within the protein–ligand complexes. Key interaction types, including, hydrogen bonds, water bridges, hydrophobic contacts, and ionic interactions, are summarized in **Fig 10**. The wild-type ApoE exhibited stable hydrogen bonds with THR108 and ASN125, along with moderate water bridge formation (e.g., ARG79, THR108, GLN128) and limited hydrophobic and ionic interactions (**Fig 10A**). The L107P mutant showed increased hydrogen bonding, particularly at THR108 and ASN125, and enhanced

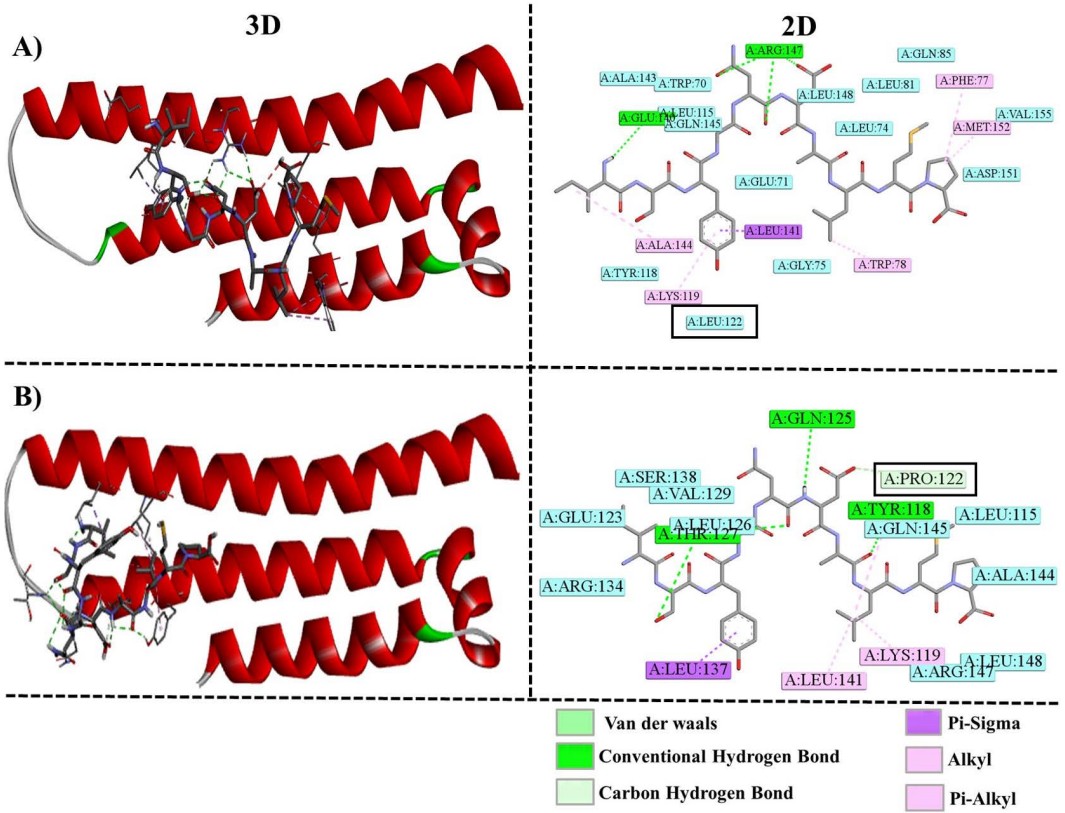

**Fig 8. Illustrates the molecular docking results comparing the interactions of native ApoE and the mutant ApoE (L122P) with Aβ.** Panel (A) depicts the 3D and 2D interaction of WT ApoE bound to Aβ. In the right-hand corner, the stick structure represents Aβ interacting with the active site residue of WT ApoE. (B) highlights the 3D and 2D interaction between the mutant ApoE (L122P) and Aβ. In the right-hand down corner, the stick structure represents Aβ interacting with the active site residue of mutant ApoE (L122P).

water bridge formation at C-terminal residues, with a slight increase in ionic interactions (**Fig 10B**). In the L122P variant, interaction patterns were further altered, displaying enhanced hydrogen bonding (including new contacts), increased hydrophobic and ionic interactions at GLU124, PHE78, and PRO122, and a distinct water bridge profile compared to other variants (**Fig 10C**). These changes suggest both mutations influence the binding interface and complex stability.

In summary, molecular dynamics simulations revealed distinct structural and interaction profiles for the L107P and L122P ApoE variants. The L107P mutation induced significant conformational instability, increased residue flexibility, structural expansion, and reduced solvent exposure. In contrast, the L122P variant demonstrated enhanced structural stability, compactness, and favorable interaction dynamics with Aβ, including increased hydrogen bonding, hydrophobic contacts, and solvent accessibility. These findings suggest that L122P may promote a stable yet potentially pathogenic ApoE–Aβ complex, while L107P likely disrupts normal protein function through structural destabilization.

## Discussion

This study employed an integrated in silico framework to systematically identify and evaluate potentially deleterious non-synonymous SNPs (nsSNPs) in the human ApoE gene, focusing on their structural, functional, and pathological relevance to Alzheimer's disease (AD). From a pool of 376 missense variants, 10 nsSNPs were consistently predicted to be deleterious by eight independent functional prediction algorithms and were therefore classified as common deleterious

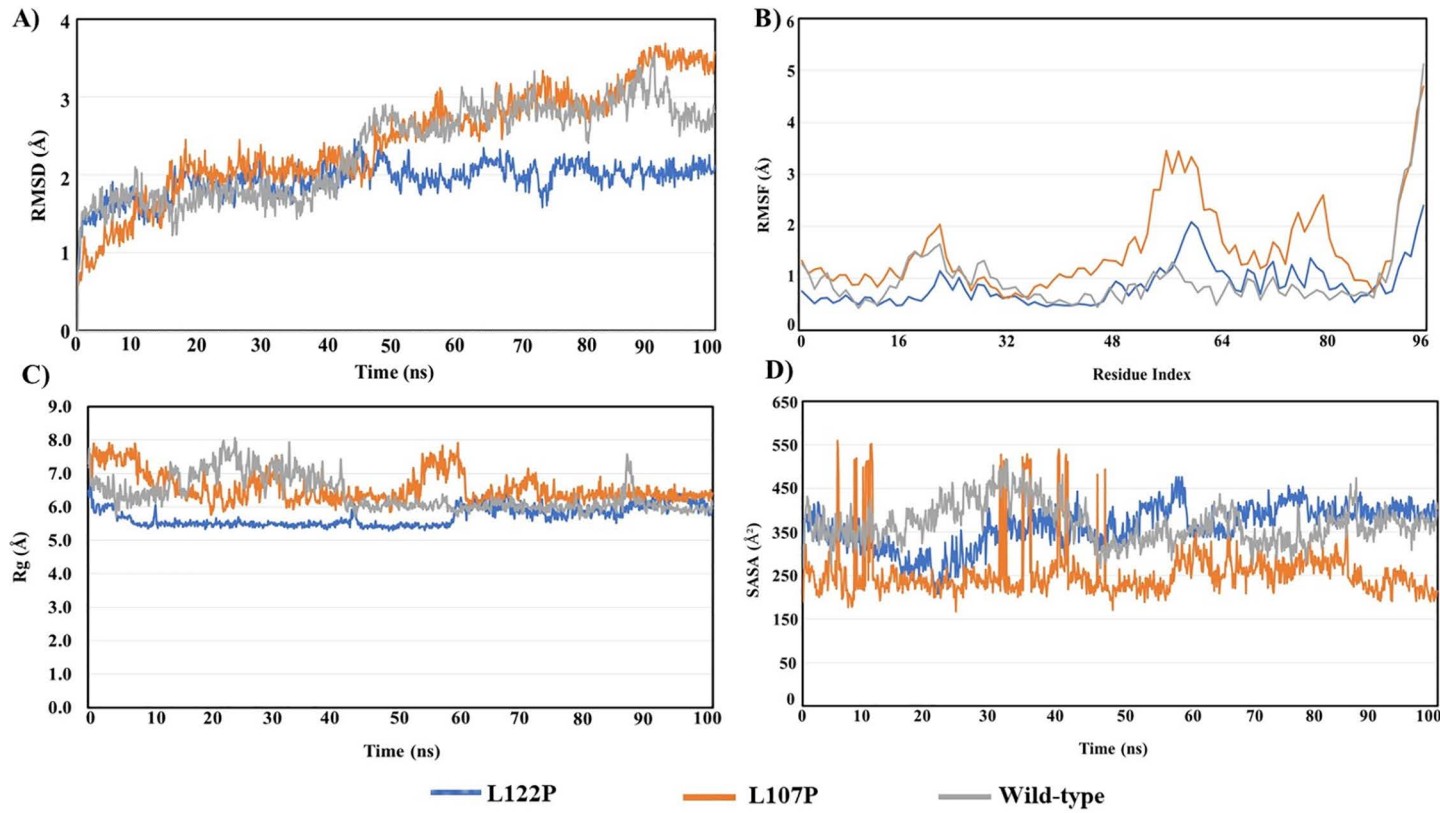

**Fig 9. Molecular dynamics simulation of WT and Mutant ApoE–Aβ complexes with a 100 ns runtime. A)** Root Mean Square Deviation (RMSD); **B)** Root Mean Square Fluctuation (RMSF); **C)** Radius of Gyration (Rg); and **D)** Solvent accessible surface area (SASA).

variants. Among these, L122P (rs1238105907) and L107P (rs1969853804) were prioritized for detailed stability and structural analyses. These variants were consistently identified by five protein stability prediction tools and four structural impact predictors as significantly destabilizing the ApoE protein, inducing substantial conformational alterations, and were therefore classified as high-risk variants. Both mutations are located within the receptor-binding domain (residues 1–167) of ApoE, a region critical for interaction with low-density lipoprotein receptor-related proteins (e.g., LRP1 and LRP8) and the amyloid precursor protein (APP)—interactions essential for the regulation of amyloid-β (Aβ) transport and clearance [6,70]. ΔΔG (delta delta G) represents the change in Gibbs free energy resulting from a mutation and serves as a key thermodynamic parameter for evaluating its impact on protein stability. A positive ΔΔG value indicates a stabilizing effect, whereas a negative value suggests destabilization [71]. In this study, the L107P and L122P mutations consistently exhibited negative ΔΔG values (ΔΔG < 0 kcal/mol) across multiple predictive tools, including I-Mutant 2.0, iStable, MUpro, INPS-MD, and DynaMut2, collectively indicating a reduction in ApoE protein stability.

. Analysis using MutPred2 revealed that the L122P and L107P nsSNPs had g-scores greater than 0.5, indicating a high likelihood of being harmful.

In comparison to wild-type ApoE, the mutant variants L122P and L107P exhibit a loss of two hydrogen bonds. Hydrogen bonds are crucial for maintaining protein structure and stability. Loss of hydrogen bonds can lead to protein denaturation, causing a loss of function [72]. Furthermore, the HOPE server predicted that the non-polar amino acid substitutions L122P and L107P primarily influence size, charge, and hydrogen bonding, potentially altering ApoE stability and interactions. In L122P and L107P, leucine is substituted with proline. Proline is smaller than leucine, and its cyclic side

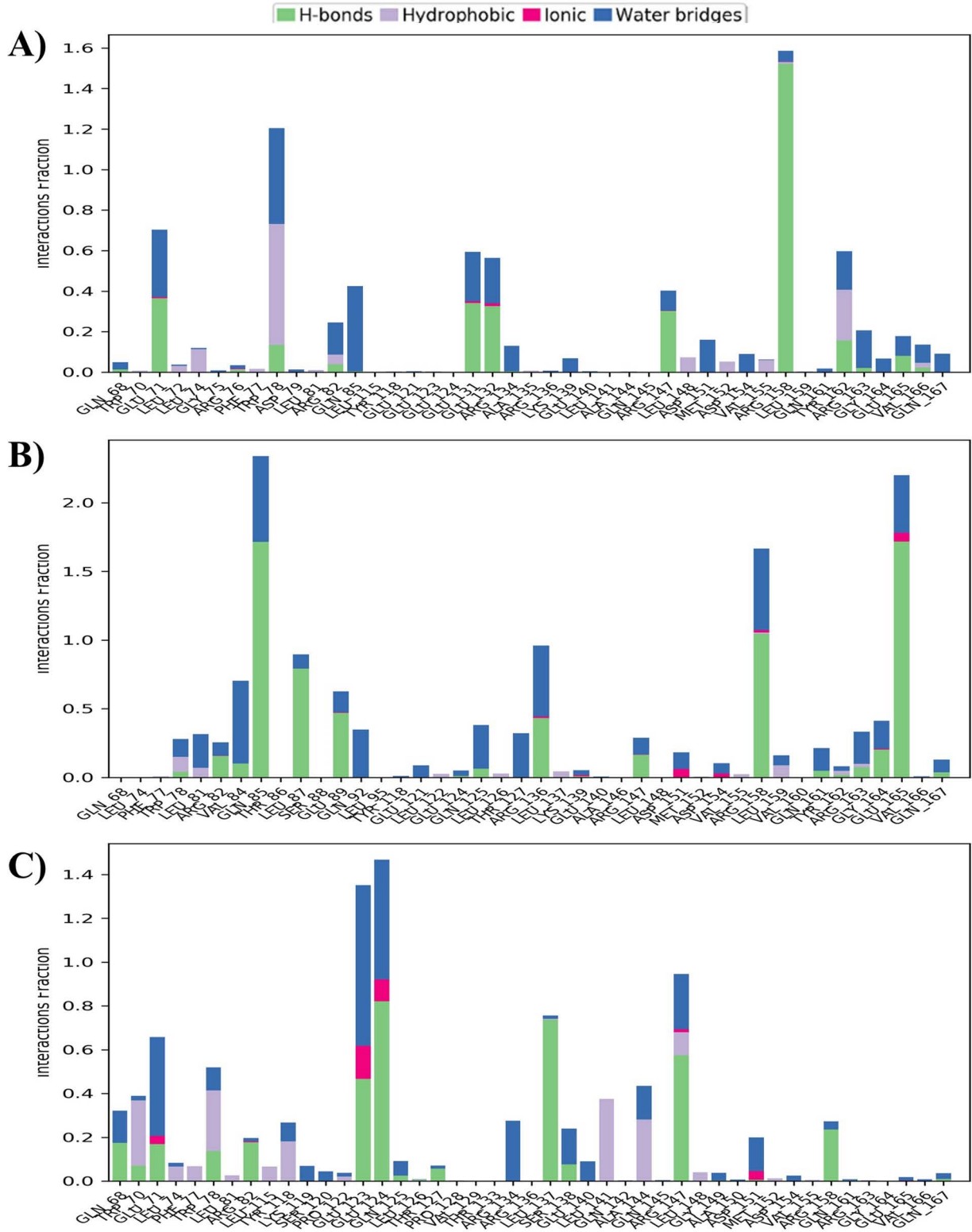

**Fig 10. Protein-ligand interaction profiles during molecular dynamics (MD) simulations. A)** WT_Aβ; **B)** L107P_Aβ; **C)** L122P_Aβ. The stacked bar charts represent the fraction of different interaction types between the ligand and residues of the protein across the MD trajectories: hydrogen bonds (green), hydrophobic interactions (purple), ionic interactions (magenta), and water bridges (blue).

chain limits conformational flexibility. Unlike leucine, proline has a rigid cyclic structure that introduces kinks into α-helices, disrupts backbone hydrogen bonding, and destabilizes local secondary structures [73]. These structural distortions expose normally buried hydrophobic residues, promoting non-specific interactions and enhancing Aβ binding. Such mutations significantly impact protein stability, often leading to denaturation and loss of function, thereby contributing to disease progression.

The receptor-binding domain of ApoE is critically involved in the regulation of Aβ metabolism, aggregation, and clearance key processes in the pathogenesis of AD [74]. Our results showed two high-risk nsSNPs, (L122P) and (L107P), within this receptor binding domain suggests that these variants might predicted to impair ApoE's normal function. Their location in this domain could influence ApoE-Aβ interactions, likely contributing to inefficient Aβ clearance and increased aggregation, thereby elevating AD risk.

Cytoscape provides biological context by enabling visualization and analysis of molecular interaction networks, helping identify key genes, pathways, and functional modules in complex systems. Our Cytoscape analysis identified ApoE as a central hub protein, exhibiting strong interactions with multiple key proteins, including CLU, APOB, APP, APOC2, APOC1, APOA1, LRP1, LRP8, APOA2, and APOC3. Notably, ApoE showed substantial connectivity with Aβ-associated proteins such as APP, CLU, and LRP1, characterized by high betweenness and closeness centrality scores. These interactions are critical for signalling cascades, amyloid precursor protein (APP) processing, and neuroprotective functions [75,76]. The receptor-binding domain of ApoE plays a pivotal role in mediating APP metabolism and Aβ clearance, primarily through interactions with LRP1 and LRP8, which facilitate neuronal lipid uptake and regulate amyloid burden [74,75]. Structural alterations in ApoE's receptor-binding domain, such as those resulting from the L122P or L107P variants, may disrupt its interaction network, potentially impairing Aβ homeostasis and diminishing neuroprotective capacity, thereby contributing to neurodegenerative disease mechanisms.

Due to the paucity of crystal structures in the protein database, SWISS MODEL was used to build the three-dimensional structures of wild-type and mutant ApoE variants. After that, ProSA-web, QMEAN, ERRAT, and Ramachandran plot analysis validated the structures. Mutations in a protein's DNA sequence can change the amino acid composition, which can alter the protein's flexibility, potentially allowing it to adopt a new conformation more suitable for binding to a specific ligand or another protein, occasionally increasing its binding affinity [57]. Structural modeling was conducted using the ApoE4 isoform, the most prevalent variant. ApoE4, characterized by arginine at positions 112 and 158, exhibits distinct domain interactions that promote amyloid aggregation and reduce clearance [13]. Using this isoform allows for focused analysis while avoiding confounding effects from the ApoE4-specific domain–domain interaction.

The active site of the protein was identified using the CASTpFold server, which revealed an increased surface area in the ApoE (L122P) mutant compared to the wild-type (WT) ApoE. This expansion of the active site surface suggests a more exposed or open binding pocket, potentially facilitating a wider range of molecular interactions [77]. Additionally, it could impact binding specificity, possibly allowing abnormal interactions or diminishing the protein's functional precision. In the case of ApoE and AD, such alterations might disrupt normal Aβ clearance mechanisms or promote the formation of pathogenic aggregates. The active site amino acid residue plays a considerable role in binding and catalytic activity in the active site of the WT and mutant ApoE protein. Among two nsSNPs, ApoE (L122P) significantly increased the binding affinity with the Aβ peptide (−6.6 kcal/mol) compared to the WT (−5.5 kcal/mol). Previous evidence suggests that the interaction between wild-type ApoE and amyloid-beta (Aβ) regulates both Aβ aggregation and clearance, thereby directly influencing amyloid plaque formation and plays a key role in AD pathogenesis [8]. Stronger binding affinity of mutant ApoE (L122P) strongly disrupt normal Aβ clearance mechanisms, potentially enhancing Aβ aggregation and promoting extracellular plaque deposition, thus strongly contributing AD pathogenesis.

The analysis of molecular dynamics simulations (MDS) offers a comprehensive understanding of the dynamic behaviour and stability of protein-ligand complexes: WT (ApoE)_Aβ, ApoE (L107P)_Aβ, and ApoE (L122P)_Aβ. These simulations provide valuable insights into various parameters such as RMSD, RMSF, RG, and SASA, shedding light on

the structural dynamics and interactions within the complexes. The RMSD values, indicative of structural deviation during the simulation period, ranged from 1.5 Å to 4 Å for the protein-ligand complexes, suggesting minimal structural changes. Notably, ApoE (L122P)_Aβ exhibited the lowest RMSD values, indicating excellent structural preservation within the complexes. Conversely, WT (ApoE)_Aβ and ApoE (L107P)_Aβ displayed slightly higher RMSD values, suggesting relatively more significant structural deviations [78–80].

The RMSF analysis provided insights into the flexibility of individual atoms within the ligands [81]. ApoE (L122P)_Aβ displayed minimal atom-level fluctuation, while WT (ApoE)_Aβ and ApoE (L107P)_Aβ exhibited greater flexibility, indicating greater fluctuation of atoms within the ligands. Regarding the radius of gyration (RG) [82], WT (ApoE)_Aβ and ApoE (L107P)_Aβ exhibited the highest value, suggesting a more extended conformation, while ApoE (L122P)_Aβ demonstrated the lowest RG value, indicating a more compact structure within the complexes. Solvent-accessible surface area (SASA) values reflected the accessibility of solvent molecules to the ligand surface, with WT (ApoE)_Aβ and ApoE (L107P)_Aβ exhibiting relatively lower solvent exposure and ApoE (L122P)_Aβ showing greater solvent accessibility. After a comprehensive simulation analysis, ApoE (L122P)_Aβ promotes a more compact and stable protein conformation, compared to WT (ApoE)_Aβ and ApoE (L107P)_Aβ.

While our predictions were supported by a combination of complementary computational approaches, including stability assessments (I-Mutant 2.0 MUpro, INPS-MD, iStable, DynaMut2), structural analyses (HOPE, MutPred2, Swiss PDB-viewer, Missense3D), and molecular dynamics simulations, it is essential to acknowledge the inherent limitations of in silico modeling. These findings, although informative and suggestive of functionally significant effects, remain speculative in the absence of experimental validation. Thus, while these results offer valuable hypotheses, they must be interpreted cautiously and validated experimentally.

Our findings align with growing evidence that rare or previously uncharacterized nsSNPs in ApoE may influence AD risk through functionally significant structural changes. Variants such as R176C and C130R, located within the receptor-binding domain, have been shown to disrupt lipidation and receptor interactions, thereby contributing to AD-related pathology [83,84]. Similarly, our identification of the L122P high-risk variant, also within this domain, with increased Aβ binding affinity, suggests its potential role in modulating ApoE function and contributing to AD pathology. These results highlight the need to investigate rare ApoE variants as contributors to disease heterogeneity and as potential targets for precision diagnostics and therapeutics aimed at reducing ApoE–Aβ interactions and enhancing Aβ clearance. Overall, this *in-silico* analysis enhances our understanding of how ApoE nsSNPs L122P may influence Aβ pathology and the progression of Alzheimer's disease. While the findings offer valuable hypotheses for future experimental validation, they should be interpreted as associative rather than causative.

## Conclusion

This study presents the first comprehensive *in-silico* characterization of L122P and L107P ApoE variants, revealing their significant structural and binding alterations with Aβ, and their possible contribution to Alzheimer's disease progression. Among them, L122P emerged as a particularly high-risk variant, showing increased binding affinity for Aβ and substantial alterations in protein conformation, solvent accessibility, and interaction dynamics. Molecular docking and dynamics simulations revealed that the leucine-to-proline substitution at position 122 introduces a novel binding interface, likely due to the rigidity of proline's cyclic structure, which disrupts local secondary motifs and exposes hydrophobic surfaces that favor Aβ interaction. This altered binding is predicted to impairs Aβ clearance, a core pathological mechanism in AD. This variant could potentially serve as novel diagnostic biomarkers for early identification of individuals at increased risk for AD, particularly if validated in future genetic association studies. Future research should also investigate the isoform-specific behavior of these variants, particularly in the ApoE4 background, to fully elucidate their role in AD progression. Overall, this study reports newly identified and previously uncharacterized high-risk ApoE variant, L122P, contributing to altered Aβ binding and structural destabilization, and offers insights for precision-medicine approaches targeting variant-specific mechanisms in Alzheimer's disease.

## Supporting information

**S1 File.** **S1 Table**. Common deleterious nsSNPs identified by eight *in-silico* tools.**S2 Table**. Structural effect of 10 common deleterious nsSNPs over ApoE protein using Missense3D tool. **S3 Table**. MutPred2 analysis of 10 common deleterious nsSNPs identified in ApoE. **S4 Table**. Structural impact of 10 common deleterious nsSNPs on ApoE protein features predicted by HOPE. **S5 Table**. Active site residue, Surface area and Volume of WT ApoE, ApoE (L122P) and ApoE (L107P). **S1 Fig:** Validation of WT and mutant ApoE model by Ramachandran Plot. a) WT ApoE, b) ApoE (L122P) and c) ApoE (L107P).
(DOCX)

## Author contributions

**Conceptualization:** Abu Zaffar Shibly.

**Data curation:** Md. Mainuddin Hossain, Juthi Adhikari, Amit Dutta.

**Formal analysis:** Md. Mainuddin Hossain, Juthi Adhikari, Amit Dutta.

**Investigation:** Md. Mainuddin Hossain, Juthi Adhikari, Amit Dutta, Abu Zaffar Shibly.

**Methodology:** Md. Mainuddin Hossain, Juthi Adhikari, Amit Dutta, Afia Khandaker, Sirajul Islam, Abu Zaffar Shibly.

**Project administration:** Juthi Adhikari, Abu Zaffar Shibly.

**Resources:** Md. Mainuddin Hossain, Juthi Adhikari, Amit Dutta.

**Software:** Md. Mainuddin Hossain, Juthi Adhikari, Amit Dutta.

**Supervision:** Abu Zaffar Shibly.

**Validation:** Md Masuder Rahman, Abu Zaffar Shibly.

**Visualization:** Md. Mainuddin Hossain, Juthi Adhikari, Amit Dutta, Abu Zaffar Shibly.

**Writing – original draft:** Md. Mainuddin Hossain, Juthi Adhikari, Amit Dutta, Abu Zaffar Shibly.

**Writing – review & editing:** Juthi Adhikari, Amit Dutta, Sirajul Islam, Md Masuder Rahman, Abu Zaffar Shibly.

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
