## [Decision Letter · Decision Letter 0]

25 Jun 2025

Dear Dr. Shibly,

Thank you for submitting your manuscript to PLOS ONE. After careful consideration, we feel that it has merit but does not fully meet PLOS ONE’s publication criteria as it currently stands. Therefore, we invite you to submit a revised version of the manuscript that addresses the points raised during the review process.

**ACADEMIC EDITOR:**

After carefully considering the reviewers comments and assessing your manuscript, I am pleased to inform you that we would like to invite you to revise and resubmit manuscript for further consideration. The reviewers have provided constructive comments that will help strengthen your work. Please address each of these points thoroughly in your revised manuscript. Additionally, ensure that you provide a detailed response letter outlining how you have addressed each comment raised by the reviewers. This will help the reviewers and myself to evaluate the changes made to the manuscript.  Any additional references suggested during the peer-review process should only be included if the authors agree that they are relevant and useful. Good luck

We look forward to receiving your revised manuscript.

Kind regards,

Khalid Raza, PhD (Computational Biology)

Academic Editor

PLOS ONE

Journal Requirements:

Reviewers' comments:

Reviewer's Responses to Questions

**Comments to the Author**

1. Is the manuscript technically sound, and do the data support the conclusions?

Reviewer #1: Yes

Reviewer #2: Yes

Reviewer #3: Yes

2. Has the statistical analysis been performed appropriately and rigorously?

Reviewer #1: Yes

Reviewer #2: Yes

Reviewer #3: N/A

3. Have the authors made all data underlying the findings in their manuscript fully available?

Reviewer #1: Yes

Reviewer #2: Yes

Reviewer #3: Yes

4. Is the manuscript presented in an intelligible fashion and written in standard English?

Reviewer #1: Yes

Reviewer #2: Yes

Reviewer #3: Yes

Reviewer #1: General comments:

The uploaded figures are mainly low resolution and not vivid enough. Make sure to update the figures with the high-resolution version later on.

Specific Comments:

Material and Methods:

- There are quite significant numbers of different isoforms for ApoE. You can check it in here:

https://www.ncbi.nlm.nih.gov/protein/?term=apolipoprotein+E+isoform. What is your consideration and justification to pick NP_001289617.1 ?

- Table 1: Kindly mention that all the deployed programs were for benchmarking purposes

- Table 1: Only PredictSNP and SNAP2 that got accuracy rate above 80%. Why not use only them for the computational works?

Homology modelling, validation and Molecular Docking study:

- How did you validate your docking protocol? What standard and/or method that you use?

- How will you determine the binding of the A-beta peptide to the ApoE protein? How to validate the binding?

- Please use online molecular dynamics application for your best protein complex to determine its stability. You can use this online dynamics to proceed https://simlab.uams.edu/ .Count all necessary parameters such as RMSD and RMSF, and use simulation time of minimum 15 ns. Alternatively, you can use CABFLEX2 to check only the RMSF in here http://212.87.3.12/CABSflex2/index

Note that I'm not the developer of those applications, nor their collaborator.

Result:

- Table 3: What can you infer from this benchmarking table? Which one is the best in terms of time and/or space complexity? Which one is the most suitable for providing biological insight? Kindly narrate accordingly!

- Kindly mention the CASTpFOLD tool-related pipeline in the methodology!

Reviewer #2: . However, several areas require clarification and refinement to enhance the manuscript’s clarity, precision, and scientific rigor.

Abstract

1. The abstract provides a comprehensive summary, but it can be made more concise. For example, replace “forecast their potential impact” with “predict their impact.”

2. Revise the aim statement for precision:

Suggested: "This study aimed to identify functionally damaging nsSNPs in the ApoE gene using in silico tools and to assess their structural and binding effects on amyloid-beta (Aβ) in the context of Alzheimer’s disease progression."

3. Emphasize the significance of the receptor-binding domain by briefly noting its importance in mediating ApoE-Aβ interaction.

4. Explain the biological implication of the L122P mutation (e.g., stronger binding affinity may promote Aβ aggregation or hinder clearance, potentially contributing to disease severity).

Introduction

1. Add a brief comparison with previously characterized pathogenic ApoE variants (e.g., C112R, R158C) to provide context and highlight novelty.

2. Elaborate on how missense mutations in ApoE disrupt receptor binding or lipid transport, which are essential for Aβ metabolism and clearance.

3. Improve flow by separating the general background from the study objective:

Suggested:

"Given the limitations of experimental approaches, this study employed in silico tools to identify deleterious nsSNPs in ApoE and evaluate their potential structural and functional effects on Aβ interaction."

4. Eliminate redundant phrasing related to Aβ accumulation and its role in AD (lines 2–7). For example, “Aβ peptide buildup and aggregation in the extracellular region” and “progressive buildup and aggregation in the brain” convey similar meanings and can be merged more concisely.

5. Add the chromosomal location of ApoE (19q13.32) and include key SNPs (rs429358 and rs7412) that define ApoE2, E3, and E4 isoforms.

6. Improve the transition from general SNP discussion to ApoE-specific focus.

Suggested: "Among genes linked to neurodegeneration, ApoE stands out due to its central role in Alzheimer’s disease, especially through structural changes introduced by specific nsSNPs."

Methodology

1. Clearly state whether population-level association data (e.g., allele frequency or GWAS statistics) are included. If not, mention this limitation.

2. Describe the criteria for identifying and selecting nsSNPs (e.g., ClinVar significance, allele frequency thresholds).

3. Explain why each computational tool was chosen, emphasizing their complementarity (sequence-based vs. structure-based) and reliability.

4. Consider incorporating DynaMut to analyze mutation-induced changes in protein flexibility and entropy.

5. Include URLs for all tools used to improve reproducibility (e.g., SIFT: http://sift.jcvi.org/).

Results

1. Provide a rationale for merging SNPs (e.g., sequence redundancy, variant clustering, or updated annotations).

2. Define and standardize terminology such as “deleterious,” “damaging,” or “high-risk” based on prediction thresholds or consensus across tools.

3. Highlight whether the 10 predicted nsSNPs lie within functional domains of ApoE (e.g., receptor-binding or lipid-interaction regions).

4. Briefly describe what Table 3 shows and how it supports the study’s conclusions.

5. Ensure all tables and figures are cited sequentially and consistently (e.g., Table 3, S3 Table).

6. If available, compare findings to previously published experimental or computational results.

7. Provide UniProt or PDB ID of the ApoE structure/template used for homology modeling in SWISS-MODEL.

8. The L122P mutation shows higher Aβ binding affinity (ΔG = –6.6 kcal/mol). Discuss its functional implications, such as gain-of-function effects that may facilitate Aβ aggregation.

9. Clarify the genetic background of the modeled ApoE sequence, specifically, whether it corresponds to ApoE2, E3, or E4 isoforms. This is essential for interpreting pathogenic relevance.

Discussion

1. Explicitly state which ApoE isoform (E3 or E4) was modeled, as isoform context affects both structure and Aβ binding behavior.

2. Justify the selection of L122P and L107P by indicating their position within functionally significant regions.

3. Provide biological context for Cytoscape network results, e.g., how ApoE mutations may affect interactions with APP or LRP1.

4. Acknowledge the speculative nature of conclusions drawn from in silico predictions. Use cautious language (e.g., “predicted to impair” instead of “impairs”).

5. Explain how proline substitutions (e.g., at position 122) may disrupt local structural motifs, exposing hydrophobic regions that promote Aβ interaction and aggregation.

6. Include a brief comparison with existing literature on rare or emerging ApoE variants (e.g., R176C, C130R) to validate findings or identify novel trends.

Conclusion

1. Emphasize novelty: state whether L122P and L107P are newly reported or previously uncharacterized high-risk variants.

2. Mention potential clinical relevance, use of findings for diagnostic markers or therapeutic targets.

3. Highlight any novel binding mechanisms or interactions uncovered during docking.

4. Begin with a clear, impactful summary of the core discovery.

Suggested: "This study presents the first comprehensive in silico characterization of L122P and L107P ApoE variants, revealing their significant structural and binding alterations with Aβ, and their possible contribution to Alzheimer’s disease progression."

5. Minimize vague expressions like “may impair” unless essential. Instead, use confident but appropriate language supported by predictions (e.g., “is predicted to impair Aβ clearance”).

Reviewer #3: The manuscript under review investigates structural alterations in Apolipoprotein E (ApoE) that impair amyloid-beta (Aβ) clearance and are associated with the pathogenesis of Alzheimer’s disease (AD). From a total of 381 non-synonymous single nucleotide polymorphisms (nsSNPs), ten were predicted to be deleterious. Among these, two nsSNPs located in the receptor-binding domain were classified as high-risk mutations. The authors analyzed the binding affinities between wild-type (WT) and mutant ApoE with the Aβ peptide, noting that the L122P mutation exhibited a higher binding affinity—suggesting its potential involvement in AD progression.

While the conceptual framework of the study is noteworthy, I do not recommend this manuscript for publication in its current form. It requires substantial revisions to improve scientific rigor, clarity, data interpretation, and overall presentation. Specific comments are outlined below:

Major Comments

1. The current title is overly generic. It should be rephrased to better reflect the study’s central hypothesis and mechanistic approach. Similarly, the keywords require revision to include relevant technical terms that capture the essence of the work.

2. The abstract suffers from grammatical issues and poorly structured sentences. It is heavily loaded with results but lacks coherence and a logical flow. Importantly, there is no concluding remark or forward-looking statement. The parenthetical phrase "excluding five" is unclear and unnecessary—please clarify its relevance or remove it.

3. The introduction presents a very basic overview and fails to build a strong scientific context. It should be rewritten to include the genetic and molecular mechanisms underlying ApoE’s role in AD. A compelling introduction must engage the reader, present relevant background, and clearly articulate the study’s rationale. Additionally, recent statistics and literature that support the gene-disease connection should be incorporated to establish relevance.

4. Methodology: This section requires significant enhancement. The SNP retrieval process should be validated against other reputable databases. Clarify what is meant by "common deleterious nsSNPs in ApoE." The heading of Table 1 should be revised for clarity, and the inclusion of input parameters in the table must be justified. Also, the rationale for using SWISS-MODEL for homology modeling should be clearly explained. Why was this tool chosen over more robust alternatives?

5. Results: The results section lacks depth and needs a more detailed discussion. In Table 3, the footnote should be expanded to explain the data presented. Elaborate on domain identification in ApoE and the implications of locating high-risk nsSNPs within these domains. The biological relevance of the protein-protein interaction findings must be thoroughly discussed. The linkage between different outputs in Table 4 needs clarification. Justify the inclusion of molecular docking and explain the impact of the L122P mutation—especially the claim regarding active site changes. In Table 5, 2D interactions should be labeled clearly, including the nature of bonding (e.g., hydrogen bonding, hydrophobic interactions).

6. Discussion: This section largely reiterates the results without offering critical analysis. The discussion should interpret findings in light of existing literature and proposed mechanisms. Potential limitations of the study should be acknowledged. Emphasize that the findings demonstrate associations and do not establish causality. Broaden the discussion to include how these results fit into larger biological or disease models.

7. The conclusion is repetitive and lacks impact. Provide a concise summary of the major findings, followed by a forward-looking statement suggesting future directions or potential applications of the research.

8. References: Many references are outdated. Replace or supplement these with more recent studies published within the last five years to ensure the literature review is current and relevant.

Minor Comments

9. Language and Grammar: The manuscript requires thorough proofreading and professional language editing. Attention should be given to sentence structure, grammar, and consistency in scientific terminology to enhance readability.

10. Figures and Schematic Diagrams: The workflow diagram appears basic and visually unappealing. Consider redesigning it using graphical elements that enhance comprehension and aesthetic quality. For inspiration, refer to workflows presented in the following publications:

DOI: 10.1021/acsomega.2c04871

DOI: 10.1021/acsptsci.2c00212

DOI: 10.1021/acs.jcim.0c00488

11. Figures and Visual Quality: Ensure all figures are of high resolution and stylistically consistent. Figure legends should be self-explanatory, clearly describing all elements shown, and aiding interpretation without requiring reference to the main text.

**Do you want your identity to be public for this peer review?** For information about this choice, including consent withdrawal, please see our Privacy Policy

Reviewer #1: No

Reviewer #2: No

Reviewer #3: No

---

## [Author Response · Author response to Decision Letter 1]

22 Jul 2025

We sincerely thank the reviewers and editor for their thorough evaluation and constructive feedback. We have carefully addressed each specific comment point by point in the response document, incorporating all suggested revisions into the manuscript to improve its clarity, scientific rigor, and overall quality. All changes have been clearly highlighted in the revised version of the manuscript. We appreciate the opportunity to revise and resubmit our work and look forward to your further consideration.

---

## [Decision Letter · Decision Letter 1]

14 Aug 2025

Computational Prediction of High-Risk Non-Synonymous SNPs in Human ApoE and 2 Their Structural Impact on Amyloid-β Interaction in Alzheimer’s Disease Pathogenesis

PONE-D-25-29477R1

Dear Dr. Shibly,

We’re pleased to inform you that your manuscript has been judged scientifically suitable for publication and will be formally accepted for publication once it meets all outstanding technical requirements.

Kind regards,

Khalid Raza, PhD (Computational Biology)

Academic Editor

PLOS ONE

Additional Editor Comments (optional):

I am pleased to inform you that your paper has been accepted for publication in PLoS ONE. Your manuscript has undergone rigorous peer review, and I am delighted to say that it has been met with praise from reviewers and editorial team. Your research makes a significant contribution to the field, and we believe it will be of great interest to our readership. On behalf of the editorial board, I extend our warmest congratulations to you.

Reviewers' comments:

Reviewer's Responses to Questions

**Comments to the Author**

Reviewer #2: All comments have been addressed

2. Is the manuscript technically sound, and do the data support the conclusions?

Reviewer #2: Yes

3. Has the statistical analysis been performed appropriately and rigorously?

Reviewer #2: N/A

4. Have the authors made all data underlying the findings in their manuscript fully available?

Reviewer #2: Yes

5. Is the manuscript presented in an intelligible fashion and written in standard English?

Reviewer #2: Yes

Reviewer #2: the authors adress all the point raised. revised manuscript is better form. Paper is acceptable in the current form.

**Do you want your identity to be public for this peer review?** For information about this choice, including consent withdrawal, please see our Privacy Policy

Reviewer #2: **Yes: ** Muhammad Saleem Khan

---

## [Editor Report · Acceptance letter]

PONE-D-25-29477R1

PLOS ONE

Dear Dr. Shibly,

I'm pleased to inform you that your manuscript has been deemed suitable for publication in PLOS ONE. Congratulations! Your manuscript is now being handed over to our production team.

Kind regards,

on behalf of

Dr. Khalid Raza

Academic Editor

PLOS ONE